


# High accuracy coastal flood mapping for Norway using LiDAR data

Kristian Breili[1,2], Matthew James Ross Simpson[1], Erlend Klokkervold[3], and Oda Roaldsdotter Ravndal[4]

[1]Geodetic Institute, Norwegian Mapping Authority, 3507 Hønefoss, Norway
[2]Faculty of Science and Technology, Norwegian University of Life Sciences, 1432 Ås, Norway
[3]Geographic Information System Development, Norwegian Mapping Authority, 3507 Hønefoss, Norway
[4]Hydrographic Service, Norwegian Mapping Authority, 4021 Stavanger, Norway

**Correspondence:** Kristian Breili (kristian.breili@kartverket.no)

**Abstract.** Using new high accuracy Light Detection and Ranging elevation data we generate coastal flooding maps for Norway. Thus far, we have mapped ∼80% of the coast, for which we currently have data of sufficient accuracy to perform our analysis. Although Norway is generally at low risk from sea-level rise largely owing to its steep topography, the maps presented here show that on local scales, many parts of the coast are potentially vulnerable to flooding. There is a considerable amount of
infrastructure at risk along the relatively long and complicated coastline. Nationwide we identify a total area of 400 km$^2$, 105,000 buildings, and 510 km of roads that are at risk of flooding from a 200 year storm-surge event at present. These numbers will increase to 610 km$^2$, 137,000, and 1340 km with projected sea-level rise to 2090 (95th percentile of RCP8.5 as recommended in planning). We find that some of our results are likely biased high owing to erroneous mapping (at least for lower water levels close to the tidal datum which delineates the coastline). A comparison of control points from different
terrain types indicates that the elevation model has a root mean square error of 0.26 m and is the largest source of uncertainty in our mapping method. The coastal flooding maps and associated statistics are freely available, and alongside the development of coastal climate services, will help communicate the risks of sea-level rise and storm surge to stakeholders. This will in turn aid coastal management and climate adaption work in Norway.

## 1   Introduction

Higher sea levels driven by anthropogenic climate change present a large challenge for many coastal communities. There are numerous negative consequences of sea-level rise, i.e., flooding, loss of life and land, damage and loss of buildings and infrastructure, increased erosion, saltwater intrusion, changing ecosystems, and reduced biodiversity (see e.g., Nicholls, 2010; Nicholls and Cazenave, 2010). The consequences of increasing sea level are large and many because the coastal zones are
densely populated areas, have a large population growth, and are economically important.

Compared to many other coastal nations, Norway is at relatively low physical vulnerability to accelerating sea-level rise (Aunan and Romstad, 2008). Norway has a very rugged coast with fjords, inlets, and many thousands of islands. The coastline



is relatively long being around 103,000 km in length (Kartverket, 2019a) and is largely characterized by steep topography and an exposed bedrock that is resistant to erosion. An important component of sea-level change for Norway is vertical land motion

(VLM) due to glacial isostatic adjustment. Regional differences in VLM essentially explain differences in observed sea-level changes along the coast. Observations from Norway's tide gauge network show that relative sea level fell over the recent period 1984-2014 around Oslo and in the middle of Norway, where VLM is largest. Whereas other parts of the coast experienced a limited sea-level rise (Breili et al., 2017). Sea level is projected to increase along the entire coastline over the 21st century albeit below the global mean change (Simpson et al., 2015, 2017). This means Norway will have to adapt to rising sea levels.

Despite these generally favorable conditions, the long and complicated nature of the coast means there are many areas, often on local scales, which are potentially vulnerable to sea-level rise. Aunan and Romstad (2008) identified three low-lying types of coastline which are at risk; (1) the strandflat which is a flattish erosional surface that fringes much of Norway; (2) glaciofluvial deltas which are often situated at the head of fjords and; (3) the soft moraine coast in the southwest of the country. Furthermore, many of Norway's cities and population centers are located on the coast and have undertaken large coastal

developments in recent years. There are also important industries (oil, fishing, aquaculture, tourism), cultural buildings, an extensive infrastructure, and many homes and cabins in the coastal zone which are potentially at risk. Cultural monuments that are close to the sea includes Bryggen, the old harbor in Bergen, and the Vega archipelago which are both on the UNESCO (United Nations Educational, Scientific and Cultural Organization) World Heritage List (UNESCO, 2019).

Coastal flooding due to storm surges have caused considerable damage along the Norwegian coast in the past. Generally

speaking, damages are limited to areas very close to the coastline owing to the steep topography. And the consequences of these extreme events have been relatively minor when compared to other parts of the world (i.e., few severe consequences like loss of life and property). There is no dataset available which allows for a complete assessment of the damage costs from these past storm-surge events. However, a sense of the costs can be understood from insurance compensation data, which give costs from building damage but not from, e.g., roads or agriculture. Insurance data from 1980 to 2018 show a total of €140 million

has been paid out owing to storm-surge damage (data from Norwegian Natural Perils Pool and Finance Norway, adjusted for inflation). The years 1987 and 2011 stand out, with annual insurance compensations of €27 and €47 million due to storm surges. Damages in 2011 were essentially caused by two storm-surge events, the storms Dagmar (24th December 2011) and Berit (26th November 2011). Given that sea levels are now rising along parts of the Norwegian coast, how might these numbers change in the future?

The consequences of future flooding can be assessed by combining sea-level scenarios with other types of geospatial data like elevation data and registers over buildings, roads, and critical infrastructure. In addition, a detailed impact assessment requires an analysis of the possibilities for adaptation, assessment of value, usage, and the expected life span of objects of impact. To our knowledge, there exists no national socioeconomic study dedicated to sea-level rise and extreme sea levels for Norway. However, Almås and Hygen (2012) looked at one aspect of the problem by examining the potential impact of

sea-level rise on Norwegian buildings. They identify approximately 110,000 buildings located within one meter above present sea level (measured in the former Norwegian height system NN1954). More than 40 percent of the buildings are anticipated of being of significant economic value, i.e., they are homes or cabins, industry, storages, hotels, restaurants, office buildings,





shops etc. Their findings indicate that the west coast of Norway is the region most at risk from sea-level rise. In total, the costs on constructional measures for adapting the existing buildings for higher sea level are estimated to €725 million. Norway is

also included as part of the European study by Vousdoukas et al. (2018a). They conclude that Norway is one of the countries that shows the highest absolute increase in expected annual damage and expected annual number of people exposed to coastal flooding towards the end of the century. They find that by 2100, annual damages will increase to between 1.7 and 5.9 percent of GDP. The main driver of this increase is climate change, with changes in economic growth patterns as a secondary effect. This result suggests that the costs of sea-level rise for Norway could be very significant.

In this study, we describe the methods and results from the first generation of nationwide inundation maps for Norway. For the first time, sea-level projections are combined with new national high accuracy Light Detection and Ranging (LiDAR) elevation data, tidal and storm-surge height information, and geospatial data in order to map coastal flooding in Norway. The main difference between our approach and past analyses is that here we use new high accuracy LiDAR elevation data. The resulting maps have been made available to end-users as part of the coastal climate service *Se havnivå i kart* (in English:

*View sea-level rise in maps*) (see Kartverket, 2019c) created by the Norwegian Mapping Authority. The service provides a web tool for visualizing the potential effects of coastal flooding and presents associated numbers over exposed objects. The demand for sea-level projections in coastal climate services is driven by three main end-user needs (Titus and Narayanan, 1995; Le Cozannet et al., 2017):

1. Identifying research needs.

2. Mitigation: To examine the consequences and benefits of sea-level projections for different greenhouse-gas emission scenarios.

3. Adaption: Understanding and communicating information that can help society adapt to present and future sea-level rise.

*Se havnivå i kart* is primarily focused on providing information that can be used in climate adaptation work. The service provides inundation maps for both present sea level and future sea level in 2090. These sea-level heights can be combined with

different return heights for storm surges which correspond to safety classes given in the current building acts and regulations for Norway (TEK, 2019). In addition, there are inundation maps indicating exposed objects and areas at 1 m height intervals set between 1 and 5 m above present Mean High Water (MHW). The service has been tailored to assist Norway's coastal municipalities with emergency preparedness, long-term planning decisions, and to help communicate the risks associated with storm surge and sea-level rise to the public and other stakeholders.

In the following we describe the methods and data used for creating the inundation maps and associated numbers over exposed objects. We show results for a variety of storm-surge heights and water levels. Furthermore, we show how these results vary regionally for different categories of land use, buildings, and roads. The discussion examines the accuracy and reliability of the maps and addresses some issues on how to interpret the results.





## 2 Methods and data

Our inundation maps and associated statistics are generated by combining sea-level projections and storm-surge return heights
with a digital elevation model (DEM) and map databases of buildings, land coverage, and roads. These data are referenced
to different vertical datums. Thus, to combine these data in a common vertical reference system requires knowledge of the
different vertical datums and how to transform between them. For example, to visualize the height of MHW in the national
height system NN2000 requires knowledge of these two vertical datums and also the relationship between them. Strauss et al.

(2012) stress that topographic vulnerability must be assessed with respect to local water levels, and not, e.g., a nationwide
definition of elevation zero. In *Se havnivå i kart*, varying tidal heights along the Norwegian coast are considered by using water
levels above MHW and storm-surge heights that include the effect of the astronomical tides. The resulting water levels can
then be transformed to the present national vertical reference system of Norway, NN2000, by exploiting MHW's known height
in NN2000.

## 2.1 Sea level projections and storm-surge return heights

Official regional sea-level projections for Norway are based on science from the Fifth Assessment Report from the Intergov-
ernmental Panel on Climate Change (IPCC AR5) (Taylor et al., 2012; Church et al., 2013). The projections show increasing
sea levels for the entire coastline over the 21st century, albeit below the global mean change (Simpson et al., 2015, 2017).
VLM due to glacial isostatic adjustment is an important component of sea-level change for Norway and observations indicate

it varies between 1 and 5 mm yr$^{-1}$ along the coast (Kierulf et al., 2014; Vestøl, 2006). VLM therefore acts to mitigate sea-level
rise in Norway and essentially explains why rates of sea-level change vary from location to location. The VLM field used in
the projections is based upon permanent GPS observations and repeated levelling (see Simpson et al., 2015). The presence of
small-scale anomalies, e.g., urban subsidence, may cause VLM to deviate significantly from this field at the local level.

Guidelines from the Norwegian Directorate for Civil Protection (DSB) recommend that the upper 95th percentile of the

spread of the projections for RCP8.5 be used in coastal planning. The upper 95th percentile corresponds to the top of the *likely
range* in IPCC terminology. As projected sea-level rise varies considerably along the coast, the projections are given for each
coastal municipality (273 in total). Depending on location, therefore, the recommended sea-level increase for use in planning
varies between 0.40 and 0.82 m (Simpson et al., 2015), see Figure 1. These numbers are rounded to the nearest 0.10 m before
use in planning.

For RCP8.5, a high emission scenario, the projected likely global temperature increase is 3-5° C for the period 2081-2100
relative to 1986-2005 (IPCC, 2013). With a view to sea-level rise, the *likely range* of the model output is considered to cover
66-100% of the total possible future outcomes (Church et al., 2013). Higher sea-level rise by 2100 can consequently not be
ruled out. There is especially large uncertainty associated with the projected contribution from the large ice sheets in Antarctica
and Greenland. Observations indicate that the ice sheet contribution has doubled since 2003 (Nicholls and Cazenave, 2010,

and references therein). DeConto and Pollard (2016) find that Antarctica has the potential to contribute more than a meter of
sea-level rise by 2100 if emissions continue unabated, but this is only one study and the physical processes required remain



controversial (see e.g., Edwards et al., 2019). We also expect further sea-level rise after 2100. Clark et al. (2016), for example, conclude that current emissions levels have committed Earth to a further global mean sea-level rise of 1.2 to 2.2 m above present sea level. While Strauss et al. (2015) find that unabated carbon emissions up to the year 2100 would lead to an eventual

global sea-level rise of 4.3 to 9.9 m. To explore sea-level scenarios above the *likely range* as recommended for coastal planning and for scenarios beyond 2100, we therefore also present numbers for water levels 1, 2, 3, 4, and 5 m above present-day MHW. These levels can help stakeholders better understand the sensitivity and vulnerability of the coast to different future scenarios.

  Storm-surge return heights are calculated from tide gauge observations, i.e., we assume no change in extreme sea levels with future climate change. Note that calculated return heights do not include the potential effects of wave setup and runup, and the

130 effects of river flooding are not explicitly included in the estimates of present and future extreme sea levels. The storm-surge return heights used here correspond to safety classes in the current building acts and regulations for Norway (TEK, 2019), i.e., water levels that on average arise once within a period of 20-, 200-, and 1000-years. The return heights were calculated by analyzing observations from 23 permanent and several hundred temporal tide gauges along the Norwegian coast (Ravndal and Sande, 2016). The Average Conditional Exceedance Rate (ACER) statistical method (Næss and Gaidai, 2009; Skjong

et al., 2013) was used, which is a type of model that allows return heights for periods longer than the tide gauge records to be estimated.

  For each coastal municipality, the return heights have been calculated for one to three locations. To be able to predict return heights at a point away from a permanent tide gauge, analysis of records from temporal tide gauges and oceanographic knowledge have been used to divide the Norwegian coast into zones with similar tidal properties. For these zones, adjusted time

series of water level can be created by first calculating the astronomical tide according to the tidal zone. Then the meteorological effect as observed by the closest permanent tide gauge is added and the ACER method is applied to the resulting time series. Unfortunately, there exist areas along the Norwegian coast where the tidal zones cannot be determined, i.e., inside fiords, bays and where narrow straits change the tidal properties over short distances. Along the southwestern coast, there is a lack of meteorological observations, and the tidal properties are complex due to an amphidromic point off the coast. The adjusted time

series for these areas are not sufficiently accurate for tidal predictions, but can still be used to calculate return heights for storm surges.

  We present maps and associated numbers visualizing both present storm-surge return heights and return heights combined with projected relative sea-level rise. The numbers for present storm-surge return heights represent today's risk and are useful for disaster preparedness, while storm-surge heights for 2090 are important for planning. Finally, in order to illustrate the

150 potential effects of a storm surge with a scenario of sea-level rise above the IPCC AR5 based projections, we include numbers for a 1000 year storm surge combined with a sea-level rise one meter above that recommended for use in planning, which may be relevant if rapid Antarctic ice mass loss becomes reality.

## 2.2 The digital elevation model

Having a DEM with high vertical accuracy and high horizontal resolution is an important prerequisite for producing reliable

inundation maps. Gesch (2009) demonstrates that high accuracy elevation data with high spatial resolution from LiDAR provide





a more accurate delineation of inundation zones than other types of elevation data. In developed areas, where small changes in the delineation of the sea may involve many objects, this can be critical.

We have used the national detailed height model of Norway (Kartverket, 2014) to estimate topographic vulnerability due to increasing sea level (available to download at www.hoydedata.no). The DEM is primarily based on airborne topographic
mapping by LiDAR but also photogrammetric matching of aerial photos of resolution 0.25×0.25 m in mountain areas. It has a spatial resolution of 1×1 m and is calculated from a point cloud of at least two points per square meter in the areas mapped by LiDAR. The vertical accuracy of the DEM has a production goal standard deviation less than 0.1 m for well-defined solid areas observed by LiDAR (Kartverket, 2014). The DEM was transformed from ellipsoidal heights to NN2000 by using the height reference surface HREF (Solheim, 2000).

Presently, about 80% of Norway is covered by the DEM, and the entire country is expected to be mapped by 2023. We assume no geomorphologic changes (e.g. erosion) or man-made landscape interventions take place over time, i.e., the same elevation data are used to map sea-level rise and storm surge today and for 2090.

In order to identify flooded zones, we have followed a "bathtub" approach similar to the one outlined in, e.g., Gesch (2009), Rowley et al. (2007), and Poulter and Halpin (2008). A particular cell in the DEM must fulfill two criteria in order to be
classified as flooded. First, it must have a height below the given sea-level rise scenario or storm-surge return height and, secondly, it must be in hydrological connection to the sea. The latter is important to eliminate low-lying areas that are protected by embankments and barriers like elevated roadbeds with heights above the sea-level scenario. The spatial extension of the sea for a given inundation level is then delineated by polygons that surround the cells in the DEM classified as flooded. Note that these polygons are not isolines with constant heights. The height of, e.g., MHW + 1 m in NN2000 varies along the coast.

## 2.3 Buildings, land cover, and road datasets

The inundation maps generated from the DEM are the basic product of *Se havnivå i kart*. Objects affected by increasing sea level can be identified by overlaying the polygons representing the flooded areas with datasets of buildings, roads, and land coverage. This approach makes it possible to map the consequences of coastal flooding for all types of geospatial data and makes the analysis more flexible than an approach where the object's height is used determine whether the object is exposed.

For roads and land coverage, we have used datasets that are customized for the scale range 1:25,000 to 1:100,000. These datasets cover the mainland of Norway and have horizontal accuracy of 2 to 50 meters. The data are cartographically edited for presentation on a scale of 1:50,000 and are named N50. To map affected buildings, the building register that is part of the Norwegian database for basic maps was used (Geonorge, 2019; Kartverket, 2019b). The datasets have a horizontal accuracy between 0.2 and 2 meters, depending on object type, location, and method used for surveying the objects. Affected buildings
are calculated by counting the number of objects inside or intersecting the polygons delineating the regions of inundation. For roads and areas, the objects are clipped, i.e., only the parts of the object inside the polygons are included in the statistics. For the roads, the length of the centerlines are summarized.

Owing to Norway's steep topography, the horizontal location is critical for determining whether an object is exposed or not. A weakness of the methodology outlined above is that objects located very close to the coast or directly above the sea surface,





e.g., buildings on piers and roads on bridges (see Figure 2), may be erroneously mapped as exposed and bias the statistics. Unfortunately, basic maps of Norway do not include attributes that allow these buildings to be sieved out and removed from the statistics.

The service's web client does not process data on the fly. All map layers and statistics are preprocessed and read from a database in order to ensure a smooth user experience. The maps are regularly updated as new knowledge and data (e.g. new

elevation data, better understanding of vertical datums, error corrections) becomes available.

## 3   Results

Inundation maps and associated statistics are presently available for approximately 80% of the Norwegian coast; see Figure 3. The maps and statistics cover the most densely populated areas and the larger coastal cities of Norway. We consider the inundation maps as the prime result of our analysis, and we first present examples of maps for geographically different areas

of Norway. We go on to present national and regional statistics for objects at risk from coastal flooding derived from the maps.

### 3.1   Examples of inundation maps of Norway

Figure 4, 5, 7, and 9 show examples of inundation maps from Smøla, Lærdalsøyri, Randaberg, and Bergen (see Figure 3 for locations). These four locations represent the three types of coastlines (glaciofluvial deltas, strandflat, and soft moraine coast) understood to be at particular risk from sea-level rise (Aunan and Romstad, 2008) and a large coastal city (Bergen). Together,

they provide examples of how different communities in Norway can be affected by coastal flooding. Four water levels are illustrated in the figures: MHW and the 200 year storm-surge level, which are mapped both for today and for 2090.

The municipality of Smøla is located on the strandflat in the middle of Norway and consists of one larger island surrounded by more than 3000 smaller islands. The strandflat is a shallow sea area with low lying land areas found typically at the mouth of fjords and along the coast between fjords. The inundation maps from Smøla, see Figure 4, indicate that sea-level rise combined

with storm surge will affect low lying coastal areas as well as piers and buildings located close to the sea. Some roads that fringe the largest island and also those which connect islands in the municipality will be flooded. The fishing village of Veiholmen (see upper part of Figure 4), located near to the northernmost part of Smøla and with a population of ∼200, appears to be at particularly high risk with many buildings adversely affected. The maps also indicate that higher sea levels may cause saline ocean water to flow into rivers and creeks, with potential effects on local ecosystems. Smøla is the municipality in Norway with

the second largest land area affected by a 200 year storm surge, both at present (1.29 m above MHW) and for 2090 (2.03 m above MHW), see Figure 13 and 14.

Figure 5 illustrates the potential effect of coastal flooding in Lærdalsøyri, a small village (population ∼1100) located on a glaciofluvial delta (see photo in Figure 6). Many glaciofluvial deltas are found at the head of fjords in Norway. These deltas are typically flat low lying areas, are densely populated, and are attractive areas for industry and businesses. As the deltas are

often surrounded by steep mountains, areas of development and agriculture are confined to the relatively flat river valley floors. In Lærdalsøyri, the combined effect of sea-level rise and a 200 year storm surge (1.00 m above present MHW, and 1.58 m





above MHW for 2090) will cause flooding in the center of the village and surrounding areas. Buildings of historic interest, government offices, industry, businesses, and some residential areas are potentially at risk. Although levees that have been built to protect the village from river flooding appear to help restrict flooding from storm surge in some areas. We find other towns
and villages located on glaciofluvial deltas show similar patterns of flooding, e.g., Lyngdal, Flåm, Fjærland, Gaupne, Stryn, Åndalsnes, Førde, Surnadalsøra, Rognan, and Alta.

While the strandflat consists of bedrock resistant to erosion, the southwest of Norway is characterized by soft sediments, sandy shores, and sand dunes (see photo in Figure 8). These areas are sparsely populated, but provide good opportunities for crop and livestock production. In general, the height of MHW is not known along the southwestern coast of Norway, except
around Randaberg illustrated in Figure 7. Despite the regions flat and low lying terrain, the inundation maps indicate only small areas affected by coastal flooding. We find similar results along the entire southwestern coast, so these areas are at low risk. However, as the shorelines largely consists of sand and soft sediments, increased erosion due to sea-level rise may become a problem for the southwest coast of Norway.

Figure 9 shows how coastal flooding will affect the city of Bergen (population ∼240,000). This municipality has the highest
number of buildings at risk from present and future coastal flooding (Figure 13 and 14). Although Bergen is characterized by steep terrain that basically prevents large areas to be flooded, the area close to the coast is densely developed. Bergen is also located in a part of Norway with a relatively high projected sea-level rise (0.71 m) as rates of glacial isostatic adjustment are lower than elsewhere, see Figure 1 and 3. The inundation maps show that projected sea-level rise alone (changes in height of MHW) will cause only small changes to the areas that will be permanently inundated. On the other hand, the combined effect
of sea-level rise and storm surges indicates many more buildings, roads, and piers will be at risk from coastal flooding in the future. Other Norwegian city centers that will become more vulnerable to coastal flooding include Fredrikstad, Sandefjord, Arendal, Mandal, Stavanger, and Tromsø. Oslo, the capital of Norway, is generally at lower risk from 21st century sea-level rise.

### 3.2 National statistics for land areas, buildings, and roads at risk

For each coastal municipality, we have calculated the area of land, number of buildings, and length of roads affected by coastal flooding. These categories are further subdivided in order to better understand the details of what is at risk, e.g., land areas are divided into areas that are developed, nature, public facility, or primary industry (see Table 1 for more details). The numbers of affected objects (i.e., nationwide totals for Norway, or at least for the ∼80% of the coast where we have data) are given in Tables 2-4 and illustrated in Figure 10. The percentage increase in exposed areas, buildings, and roads between 2017 and 2090
for different sea-level scenarios are listed in Table 5.

We first note that initial analysis of the numbers of objects affected by coastal flooding for present MHW appear to be biased high. Here we consider MHW as the water level at which objects are permanently inundated by coastal flooding. For present MHW, therefore, the numbers of affected objects should be close to zero. However, we find an area of 152 km$^2$, 40,072 buildings, and 180 km of roads mapped as permanently flooded for present MHW. The large area identified as flooded for
present MHW indicates that there is a misfit between the polygons that define the flooded area and the land tiles used in





Norwegian maps. In principle, the polygon and the coastline should match, but there are misfits due to different methods of mapping (LiDAR vs. photogrammetric analysis of aerial photos) and inaccuracies in the methods and data used in the analysis. Inspection of detailed maps and aerial photos indicates that many of the buildings erroneously mapped as flooded for present MHW are small boat houses situated very close to the coast or buildings on piers or pillars above the water surface, see Figure 2.

Roads erroneously mapped as flooded for present MHW include road sections on bridges and in underwater tunnels.

The numbers of affected objects for the storm-surge return heights (e.g., 200 year storm-surge height for 2090) are considerably higher than those for present MHW. To some extent, these numbers will also be in error owing to the present MHW bias. In order to reduce the effect of the MHW bias, we subtract the numbers calculated for present MHW for areas and roads where available. This implies that the size of affected areas is calculated between surfaces mapped with consistent methods. For roads

it is unlikely that segments on bridges and in underwater tunnels will be affected, even for higher storm-surge return heights.

We can not, however, simply subtract the numbers calculated for present MHW from the numbers for higher water levels for buildings, because an unknown number of these buildings will truly be affected by higher levels of flooding. We suggest that the numbers of buildings erroneously mapped as affected will decrease for higher water levels. The numbers calculated for present MHW for buildings form a basis estimates for other water levels can be compared to. They can also be considered as a

270 measure of the precision of the current methods and data used in our analysis. Note that because the coastal climate service *Se havnivå i kart* presents numbers including the MHW-bias, the numbers for affected areas and roads given in Table 2 and 4 will differ from those of *Se havnivå i kart*.

Our results help quantify the risk of present-day coastal flooding for Norway and how that risk will increase with sea-level rise. If we compare totals of what is exposed to a 200 year storm surge at present and for 2090, we can broadly see how that

risk will evolve nationwide. Total land area exposed will increase from around 400 to 610 km$^2$, total number of buildings from 105,000 to 137,000, and total length of roads from 510 to 1340 km. A well recognized consequence of sea-level rise is that present-day storm-surge levels will be reached or be exceeded far more frequently in the future (for Norway see Simpson et al., 2017). This is also apparent from our statistics. For example, the numbers of affected objects for the 20 year storm-surge return height in 2090 exceed the numbers for the 1000 year storm-surge height at present.

For all water levels, Table 2 indicates that the vast majority of flooded land areas fall into two categories; nature and primary industries (see Table 1 for a more detailed description of subcategories). Only a small fraction is categorized as developed or public facility. This reflects the fact that 94.8% of Norway's total land area is nature and undeveloped land areas, 3.4% is agricultural areas, and only 1.7% is developed (SSB, 2019). However, we note that developed areas exposed to a 200 year storm surge will increase by 200% in size between now and 2090 (increasing from 6 to 19 km$^2$, see Table 2 and 5). A majority of the

affected buildings are private homes and private industry, while the fraction of public buildings is small (see Table 3). We have also identified some exposed buildings categorized as critical infrastructure (see Table 1 for definition). These buildings must function during crises because their failure may cause vital public services to break down. It is therefore especially important to identify these buildings so that climate adaptation measures can be taken. Table 3 and 5 show that the number of buildings categorized as critical infrastructure at risk from coastal flooding will more than double due to projected 21st century sea-level




rise (from 30 to 80 for the 200 year storm-surge level). For roads that are exposed, there is an approximate balance between private and public roads (Table 4).

There are noticeable differences in the statistics between the different present-day storm-surge return heights (see Table 6). The increases from the 20 year to the 200 year present-day storm-surge height are 12 and 22% for the number of affected buildings and the size of flooded land areas, respectively. The increase from the 200 year to the 1000 year present-day storm-295 surge return height is 7 and 11% for buildings and areas. For roads, the increase is a lot larger, i.e., the length of roads flooded increases by 72% between the 20 year and the 200 year present-day storm-surge return heights, and by 31% from the 200-year to the 1000 year return height. Taking into account projected sea-level rise for 2090, the increases in affected objects between the different storm-surge return heights show a similar pattern to the present day. That is, the increases for higher water levels are more rapid for roads compared to buildings and land areas.

Table 2-4 also includes numbers for present MHW plus 1, 2, 3, 4, and 5 m, as well as the 1000 year storm-surge return height plus one meter for present and 2090. Global sea-level rise will continue after 2100 and these numbers are therefore of use when assessing the consequences of long-term sea-level rise. The numbers for MHW plus 5 m represent the lower limit of the range of eventual global sea-level change, suggested by Strauss et al. (2015). In this scenario, more than 1700 km$^2$, 263,000 buildings, and 6800 km of roads would be permanently flooded. The numbers for a 1000 year storm surge plus one 305 meter sea-level rise are though smaller than those for MHW+5 m, but are still significantly higher than those for 200 year and 1000 year storm surge for 2090. The consequences of long-term sea-level rise for Norway are profound, will lead to large changes to many coastal cities and to the nature of the coastline, and will require extensive climate adaptation measures.

### 3.3 Regional statistics for land areas, buildings, and roads at risk

Here we present results for each coastal municipality in Norway. Regional differences are useful for identifying areas of the 310 coast that are most vulnerable to sea-level rise and storm surges. We focus on the 200 year storm-surge return height but note that the pattern of impacts are broadly similar for other return heights. Figure 11 and 12 show for each coastal municipality the area of land, number of buildings, and length of roads that are affected by coastal flooding at present and for 2090, respectively

The municipalities with the largest land areas that are at risk of flooding are located in the middle of Norway (between Trondheim and Lofoten) and in the outer part of Oslofjorden. This is also evident by the left panels of Figure 13 and 14, which 315 summarize results for the ten municipalities that have the highest number of affected objects. For the present-day 200 year storm-surge return height, nine of these ten municipalities are located in the middle of Norway and one (Fredrikstad) in outer Oslofjorden. For 2090, the size of the flooded area increases and the order of the municipalities changes slightly, but the general pattern of regional impacts does not change.

The regional pattern of land areas affected by coastal flooding closely corresponds to regional differences in the storm-surge 320 return heights, although regional differences in topography and projected sea-level rise also plays a role to some extent. The exposure of land areas to coastal flooding is one measure of the impacts of 21st century sea-level rise and storm surge. As mentioned above, the majority (> 80%) of these land areas are classified as nature. Several of the municipalities in the middle of Norway, where the largest land areas are flooded, are sparsely populated. This is also evident from the maps visualizing the





distribution of affected buildings (middle part of Figure 11 and 12), which have quite different spatial patterns. For buildings,
the consequences of storm-surge flooding is particularly large in two counties, Hordaland and Rogaland, which are on the west
coast of Norway. Moreover, many buildings are exposed along the outer parts of Oslofjorden, along the southern coast, around
Trondheimsfjorden, in Lofoten, and in Tromsø. These regions stand out as they are densely populated and include several of
the largest cities in Norway.

The pattern of exposed roads (right panel of Figure 11 and 12) is similar to that for land areas, but the ten most exposed
municipalities also includes some locations along the southernmost part of the coast.

## 4    Discussion

### 4.1    Uncertainties of mapping

A number of different factors determine the accuracy of the inundation maps and associated statistics of exposed objects.
Although the uncertainties attached to these factors are not accounted for in our analysis, we discuss their relative importance
to the results. Factors determining the accuracy of our results include uncertainties related to (1) the DEM, (2) the vertical
reference frame NN2000, (3) the transformation of ellipsoidal heights to the national height system (HREF), (4) the height
determined for mean sea level and MHW, (5) the estimated storm-surge return heights, (6) the sea-level projections, (7) the
horizontal position of buildings and roads, (8) inaccurate polygons defining land cover, and (9) the effect of, e.g., buildings
on pillars and piers. We note that these factors and their uncertainties are inherently different. Furthermore, not all of these
factors are relevant for all of the water levels we have mapped. Uncertainties related to storm-surge heights are, for example,
not relevant when mapping MHW.

When assessing future flood risk the largest uncertainty probably relates to the sea-level projections (see Table 7). The sea-
level projections have uncertainties related to the future emission scenario and the ability of models to simulate the future
sea-level response. For the mapping method approach taken here, however, where sea-level rise is considered a fixed number
(95th percentile of RCP8.5), the uncertainty associated with the sea-level projections can be ignored. In this situation, planning
policy dictates which sea level number to use, but there will nevertheless be mapping uncertainties related to, e.g., the accuracy
of the DEM and tidal datums.

The DEM has a project goal root mean square error (RMSE) of less than 0.1 m (Kartverket, 2014). This is ensured by
comparing and fitting the point cloud of LiDAR measurements to control-fields and road tracks with heights observed by
Global Navigation Satellite Systems (GNSS). Both control-fields and road tracks must be considered as favorable LiDAR
targets. The actual accuracy of the DEM depends on the slope of the terrain, terrain surface complexity, target reflectivity,
canopy coverage and near ground vegetation, the density and distribution of the ground returns, the accuracy of the LiDAR
system, the interpolation algorithm used to create the DEM from the source data, and the spatial resolution of the DEM (e.g.,
Reutebuch et al., 2003; Li, 1992). Furthermore, transforming ellipsoidal heights to the national height system NN2000 may
introduce additional errors. As heights observed by both GNSS and LiDAR are transformed to NN2000 using the same HREF





model, any errors in the transformation will not be detected by comparison to the GNSS control measurements. We therefore consider the project goal RMSE as an optimistic error estimate for the coastal zone.

In Norway, MHW corresponds to the height of the M2 tidal constituent above mean sea level. The uncertainty of MHW therefore depends on the definition of mean sea level, the uncertainty of the estimated amplitude of M2, and the height differ-360  ence between MHW and NN2000. In addition, other tidal constituents give small contributions to the mean high tide that the present definition of MHW does not include. Unfortunately, there are no assessments of the uncertainty of MHW along the Norwegian coast. But what we can say is that the tidal datums, storm-surge levels, and their heights with respect to NN2000 are well known in areas close to the tide gauges. Along other parts of the coast, they are less well defined. Uncertainties associated with the tidal datums and storm-surge levels may therefore exceed the project goal uncertainty (RMSE<0.1 m) of the elevation 365  data in some areas.

There are also effects that are not included in our analysis; for example, wave setup and runup, changes in tides due to sea-level rise, coastal erosion, and the effects of river flooding close to the coast. We have assumed no future changes to the storm-surge return heights but note that a recent study projects areas of increase, areas of decrease, and also areas of model disagreement along the Norwegian coast (Vousdoukas et al., 2018b).

## 4.2  Accuracy of the DEM

The accuracy of a DEM can be assessed by comparing it to surveyed control points located in various types of terrain. For example, Gesch (2009) assessed elevation data over eastern North Carolina, USA, by comparing it to 489 control points the National Geological Survey uses for gravity and geoid modeling. These points were surveyed by GNSS. Poulter and Halpin (2008) followed a similar approach that also focused on North Carolina, but used 3480 quality control points surveyed by real 375  time kinematic GNSS.

We assess the quality of Norway's DEM using two independent sets of control points surveyed by GNSS. The first dataset includes about 10,000 points that are part of the Norwegian national geodetic network (NGN). These points are spread throughout Norway in various types of terrain and topography, and are in locations suitable for making GNSS measurements (i.e. sites are chosen where obstacles that could interrupt the GNSS signals are avoided). An adjustment of baselines and 3D positions 380  of individual benchmarks was used to compute final heights with typical standard errors of less than 1 cm. The second data set consists of 132 points observed with the Norwegian real time kinematic GNSS network service, known as CPOS (Ouassou et al., 2015). The test field covered an area of approximately $0.2$ km$^2$ and was located in typical Norwegian coastal terrain including solid bedrock, slopes, and beaches covered with boulders. As these points were surveyed with CPOS, we expect that the heights in the test field have some lower accuracy compared to the NGN, approximately 2 to 3 cm. All heights observed by 385  GNSS were transformed to NN2000 by use of HREF. For both groups of control points, we estimated the maximum, minimum, and mean difference as well as the RMSE. Table 8 summarizes the results of the comparison.

Comparing the nationwide DEM to the heights of the NGN reveals large differences ranging up to ∼62 m. Many of the largest differences are for control points located on the roofs of high buildings or in open-pit mines where the terrain has changed due to human activity. We therefore opt to eliminate these outliers, and focus on control points within ±1 m from the




DEM only. For these remaining points, the mean difference is -0.12 m and RMSE is 0.26 m. The negative bias indicates that
the DEM has systematically lower heights than the NGN. Benchmarks located at high points in the terrain may partly explain
this bias. For instance, many of the benchmarks in NGN are placed on small concrete pillars with horizontal dimensions of
0.5×0.5 m and a height of approximately 0.25 m above its local surrounding terrain. Such features are not picked up by the
DEM because the area of the pillars amounts to only one forth of a cell in the DEM. Also the algorithm used to convert the

LiDAR data from a point cloud to a regular grid may contribute to the bias. The generalization can be considered as applying
a low pass filter to the terrain, with the effect of filtering out the finest details in the terrain. The RMSE of 0.26 m is similar to
that calculated by Poulter and Halpin (2008) for a 6×6 m DEM covering North Carolina, but significantly higher than 0.14 m
estimated by Gesch (2009) for the 3×3 m USGS National Elevation Dataset also covering North Carolina.

Comparing the DEM to the points in the coastal test area, we calculate a mean difference of 0.11 m and RMSE of 0.28 m.

The coastal test field also has points with differences larger than one meter, these points are located in steep terrain close to the
sea. Using observations from flat terrain only, the mean difference and the RMSE reduce to -0.01 m and 0.10 m, respectively.
We repeat the calculations by replacing the nationwide DEM with a DEM with a finer spatial resolution (0.5×0.5 m) that
covers most of the test field. Using a finer spatial resolution acts to reduce the overall RMSE by 44% and several of the
largest differences also become smaller (see Table 8). This indicates that the vertical accuracy of the DEM can be significantly

improved by increasing the spatial resolution to above 1×1 m, and especially in steep terrain.

Our tests suggest that the project goal of the DEM used to calculate the inundation maps (RMSE<0.1 m) is only achieved in
flat terrain and considerably lower accuracies must be expected in steep areas and along much of the coast. The comparison to
control points in the national geodetic network indicates a RMSE of 0.26 m to be a more realistic error estimate. As the control
points in NGN are located in different types of terrain, which broadly reflect Norway's varying physical geography, we believe

they provide a more appropriate DEM quality indicator rather than comparisons to measurements at idealized control surfaces
and road tracks as used to determine the project goal RMSE.

Any error in the DEM translates into horizontal errors when mapping the extent of a water level, or flood surface. For a
particular section, the overall horizontal deformation can be written $\epsilon/\tan(\alpha)$, where $\epsilon$ is the uncertainty of the DEM and $\alpha$ is
the slope of the terrain. In steep terrain, we expect that the DEM has its largest errors, but the horizontal deformation due to a

large vertical error will be small. In flat terrain, it is opposite; the DEM is typically more accurate, but a smaller vertical error
may introduce larger horizontal deformations. For example, a DEM error of 0.26 m will deform the line that delineates a flood
surface by 2.97 m and 0.71 m for a 5 and 20° slope, respectively. From this we can summarize that, although Norway has a
generally steep coastal topography, the relative large DEM errors here will not introduce large horizontal errors when mapping
flood levels. However, given the length of the coast, and large amount of infrastructure located very close to the coastline, the

DEM errors may be critical for determining which objects are at risk.

### 4.3 Comparison to other work

As an alternative to our approach where affected objects were identified by overlaying inundation polygons with geospatial data
like buildings, the height of the objects themselves can be used to identify what is exposed to future sea-level rise and storm





surges. Almås and Hygen (2012) followed this approach and used a DEM (unknown spatial resolution but likely $10 \times 10$ m

horizontal resolution with at best 2-3 m vertical error) to determine heights of buildings in the coastal zone. In their study, approximately 110,000 buildings were found nationwide with a height less than one meter above elevation zero in the former national vertical reference system of Norway, NN1954, which at Norwegian tide gauges has its zero height within -0.09 m and 0.17 m from mean sea level. Unfortunately, a straightforward comparison of the findings of Almås and Hygen (2012) with our results (Table 3) is not possible. Firstly, this is because we have not analyzed affected buildings for a fixed height, but

have taken into account tidal variations. This will likely make a significant difference because MHW ranges from a couple of centimeters to 1.1 m above mean sea level in Norway. If not taken into account, the flooding risk will be underestimated in areas with mean high tide elevation exceeding 0 m, and comparisons across regions with different tidal levels will be compromised (Strauss et al., 2012). Secondly, we have used NN2000 as vertical reference frame instead of NN1954. At the tide gauges along the Norwegian coast, the difference between these two vertical reference frames varies between -15 cm and 12 cm (Simpson

et al., 2015). Thirdly, the numbers in Table 3 are based on data that cover 80% of the coast, while the study by Almås and Hygen (2012) covers the entire coast. If we still attempt to compare numbers, the water level MHW + 1 m is perhaps the most similar to the height used in their analysis. For MHW + 1 m our results show 86,944 affected buildings, which is significantly less than the ∼110,000 reported by Almås and Hygen (2012). Note that MHW+1 m in most areas will be higher than height 1 m in NN1954.

The present study does not aim at being a socioeconomic analysis of coastal flooding for Norway as the climate service includes no information on value of property or the population in the coastal zone. Our inundation maps, however, could be used as input to a socioeconomically analysis. In their analysis, Vousdoukas et al. (2018a) caution that the accuracy of their modeled extreme sea levels for Norway may be affected by the presence of many bays, islands and steep complex terrain. Furthermore, they indicate that elevation data of higher spatial resolution are required to achieve the same accuracy for Norway

as for flatter parts of Europe. This suggests that high accuracy national coastal flooding maps must be used to achieve results that are useful for planners and stakeholders. We believe that the methods and data used for mapping sea levels in the present study, especially the use of a $1 \times 1$ m DEM and accounting for regional differences in MHW, storm-surge heights and sea-level rise, represent significant progress compared to the methods used by Almås and Hygen (2012) and Vousdoukas et al. (2018a).

## 5  Conclusions

Using new high accuracy LiDAR elevation data we have generated coastal flooding maps for Norway. Thus far, we have mapped ∼80% of the coast, for which we currently have data of sufficient accuracy to perform our analysis. Our mapping method accounts for regional variations in tidal datums, storm-surge levels, and projections of sea-level rise. Nationwide we have identified a total area of 400 km$^2$, 105,000 buildings, and 510 km of roads that are at risk of flooding from a 200 year storm-surge event at present. These numbers will increase to 610 km$^2$, 137,000, and 1340 km with projected sea-level rise to

2090 (95th percentile of RCP8.5 as recommended in planning). If sea-level rise exceeds the projections by 1 m, then an area





of 1060 km$^2$, 189,000 buildings, and 3490 km of roads would be exposed to 1000 year storm surge. This gives an indication of how vulnerable Norway is to a scenario of rapid ice melt from Antarctica.

Examining the categories of what is at flooding risk shows the vast majority of areas are classified as nature. However, the fraction of total area classified as developed, public facility, or primary industry increases for higher water levels. Developed areas at flooding risk from a 200 year storm surge will increase more than three times in size between now and 2090 due to sea-level rise (increasing from 6 to 19 km$^2$). For buildings, around 80% of those at risk are private (homes, cabins, garages, or boat houses) for all mapped water levels. The fraction of buildings classified as private industry, public, or critical infrastructure increases for higher water levels. Critical infrastructure buildings at risk from a 200 year storm surge will increase from 30 to 80 between now and 2090. For roads, the percentage of public roads at risk will increase for higher water levels. Thus, while sea-level rise leads to more objects to be at risk of flooding, our results also indicate an increasing fraction will be objects of higher value.

Regional differences indicate that the west and southern coast of Norway, outer parts of Oslofjorden, areas around Trondheimsfjorden, and Tromsø have the largest numbers of buildings at risk of coastal flooding. For land areas and roads, it is the middle of Norway and outer Oslofjorden that are most at risk. Regional differences in the number of objects exposed to flooding can largely be explained by regional differences in population density. Inspection of the inundation maps shows that, across much of Norway, the typically steep topography restricts flooding to areas immediately adjacent to the coast. Of the examples we have examined, we find cities, island communities, and in particular towns and villages located on glaciomarine deltas are at risk from coastal flooding. The flooding risk at glaciomarine deltas can be exacerbated by the effect of river flooding.

A number of different factors determine the accuracy of the mapping and associated statistics of exposed objects. A comparison of control points from different terrain types indicates that the elevation model has a RMSE of 0.26 m and is the largest source of uncertainty in our mapping method. There are also smaller errors associated with different vertical datums and transformations between datums that have not been assessed for the entire coast. However, we believe that the sum of these mapping errors are generally smaller than the projected sea-level rise, which gives us confidence in our results. Despite the generally steep nature of the coastline, where any mapping errors introduce only small errors in the horizontal extent of flooding, the sheer length of the coast means that small errors can accumulate. A lot of infrastructure is located very close to the coast and may therefore be erroneously mapped as exposed (or not at risk). Furthermore, objects situated directly above the water surface, e.g., buildings on pillars and roads over bridges, will be erroneously mapped as exposed and cannot be sieved from our results. Owing to this, some results will be biased high. For example, we find 40,000 buildings and 180 km of roads erroneously mapped as exposed to present MHW, when the true number should be zero.

Although Norway is generally at low risk from sea-level rise largely owing to its steep topography, the maps presented here show that on local scales, many parts of the coast are potentially vulnerable to flooding. Norway is a well developed country, with expensive infrastructure, properties of high commercial value, and buildings of high standards. These factors raise the potential costs of flooding but make climate adaptation measures more cost effective. Our coastal flooding maps and associated statistics are freely available, and alongside the development of the coastal climate service *Se havnivå i kart*, will help communicate the risks of sea-level rise and storm surge to stakeholders. This will in turn aid coastal management and



climate adaption work in Norway. Users should keep in mind that our maps help identify areas of potential risk, rather than provide exact answers, and that there are uncertainties related to the mapping method and physical processes (e.g. waves) not included here. For planning decisions, a site visit and additional analysis may therefore be appropriate.

*Author contributions.* KB and MS wrote the paper, analyzed and interpreted the statistics, and made all figures, EK has done the GIS-
495 analysis and configured the system for producing inundation maps and associated statistics, OR contributed materials to the GIS-analysis, and all authors are members of the team developing and maintaining the coastal climate service *Se havnivå i kart*.

*Competing interests.* The authors declare no conflict of interest

*Acknowledgements.* The authors are thankful to Magnhild Aspevik who provided the photo from Lærdalsøyri (see Figure 6) and the development team behind the open source QGIS software used to create the presented inundation maps.





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



**Table 1.** Overview of categories and subcategories of objects potentially exposed to coastal flooding.

| Category | Sub category | Examples of object types |
|---|---|---|
| Buildings | Private | homes, cabins, garages, boat houses |
| | | factories, workshops, storage halls |
| | | power plants, transformers |
| | Private industry | agricultural buildings, fish farming facilities |
| | | offices, bank buildings, post offices, TV buildings |
| | | shopping centers, petrol stations, parking houses |
| | | hotels, restaurants, canteen buildings, rental cabins |
| | Public | administration buildings, town halls |
| | | waste handling, water supply, pump stations |
| | | railway and subway stations, freight terminals |
| | | universities, schools, student homes |
| | | galleries, libraries, sport halls, |
| | | buildings for religious activities |
| | | clinics, medical centers, living and service centers |
| | | lighthouses, monuments, public toilets |
| | Critical infrastructure | hospitals, ambulance stations, nursing homes |
| | | prisons, fire stations |
| Areas | Developed | cities, residential estates, industry areas, airports |
| | Nature | forests, wetlands, fields, glaciers |
| | Public facility | sports facilities, cemeteries |
| | Primary industry | agricultural areas, quarries |
| Roads | Private | privately owned roads |
| | Public | European routes, highways, county roads, municipality roads |





**Table 2.** Affected areas [km$^2$] in Norway under different water levels at present and for 2090. Numbers in parentheses indicate the percentage share of the total for each subcategory. Note that the areas for present MHW have been subtracted from the numbers for higher water levels.

| Scenario | Year | Total | Developed | Nature | Public facility | Primary industry |
|---|---|---|---|---|---|---|
| MHW | 2090 | 128.5 | 1.0 (0.8) | 121.0 (94.2) | 0.0 (0.0) | 6.6 (5.1) |
| 20 yr | present | 330.5 | 3.9 (1.2) | 294.6 (89.1) | 0.3 (0.1) | 31.8 (9.6) |
| 20 yr | 2090 | 530.6 | 13.6 (2.6) | 453.6 (85.5) | 1.0 (0.2) | 62.5 (11.8) |
| 200 yr | present | 402.0 | 6.4 (1.6) | 351.0 (87.3) | 0.6 (0.1) | 44.0 (11.0) |
| 200 yr | 2090 | 610.3 | 19.3 (3.2) | 514.5 (84.3) | 1.2 (0.2) | 75.3 (12.3) |
| 1000 yr | present | 446.8 | 8.5 (1.9) | 386.8 (86.6) | 0.7 (0.2) | 50.7 (11.4) |
| 1000 yr | 2090 | 660.6 | 23.6 (3.6) | 551.2 (83.4) | 1.4 (0.2) | 84.4 (12.8) |
| MHW+1 m | present | 273.9 | 3.1 (1.1) | 246.6 (90.0) | 0.2 (0.1) | 24.0 (8.8) |
| 1000 yr+1 m | present | 851.3 | 40.5 (4.8) | 687.8 (80.8) | 2.1 (0.2) | 120.8 (14.2) |
| 1000 yr+1 m | 2090 | 1056.8 | 54.0 (5.1) | 840.0 (79.5) | 2.8 (0.3) | 160.1 (15.2) |
| MHW+2 m | present | 647.2 | 24.4 (3.8) | 540.6 (83.5) | 1.5 (0.2) | 80.8 (12.5) |
| MHW+3 m | present | 1032.5 | 52.5 (5.1) | 824.3 (79.8) | 2.6 (0.3) | 153.0 (14.8) |
| MHW+4 m | present | 1379.7 | 69.8 (5.1) | 1082.8 (78.5) | 3.8 (0.3) | 223.2 (16.1) |
| MHW+5 m | present | 1719.4 | 84.4 (4.9) | 1337.7 (77.8) | 4.9 (0.3) | 292.5 (17.0) |





**Table 3.** Affected buildings in Norway under different water levels at present and for 2090. Numbers in parentheses indicate the percentage share of the total for each subcategory.

| Scenario | Year | Total | Private | Private industry | Public | Critical infrastructure |
|---|---|---|---|---|---|---|
| MHW | present | 40072 | 32677 (81.5) | 6891 (17.2) | 448 (1.1) | 6 (0.01) |
| MHW | 2090 | 61252 | 51436 (84.0) | 9122 (14.9) | 606 (1.0) | 7 (0.01) |
| 20 yr | present | 93566 | 78721 (84.1) | 13512 (14.4) | 1137 (1.2) | 22 (0.02) |
| 20 yr | 2090 | 125904 | 101665 (80.7) | 21227 (16.9) | 2457 (2.0) | 65 (0.05) |
| 200 yr | present | 105180 | 87370 (83.1) | 16007 (15.2) | 1505 (1.4) | 30 (0.03) |
| 200 yr | 2090 | 137313 | 109983 (80.1) | 23614 (17.2) | 2980 (2.2) | 80 (0.06) |
| 1000 yr | present | 112286 | 92523 (82.4) | 17623 (15.7) | 1763 (1.6) | 35 (0.03) |
| 1000 yr | 2090 | 143684 | 114359 (79.6) | 25120 (17.5) | 3362 (2.3) | 89 (0.06) |
| 1000 yr+1 m | present | 166158 | 129875 (78.2) | 30369 (18.3) | 4641 (2.8) | 151 (0.1) |
| 1000 yr+1 m | 2090 | 189155 | 147591 (78.0) | 34322 (18.1) | 5602 (3.0) | 191 (0.1) |
| MHW+1 m | present | 86944 | 72999 (84.0) | 12735 (14.6) | 1037 (1.2) | 12 (0.01) |
| MHW+2 m | present | 141649 | 112137 (79.2) | 25255 (17.8) | 3391 (2.4) | 91 (0.06) |
| MHW+3 m | present | 185175 | 144329 (77.9) | 33804 (18.3) | 5441 (2.9) | 192 (0.10) |
| MHW+4 m | present | 223396 | 175011 (78.3) | 39345 (17.6) | 6926 (3.1) | 258 (0.12) |
| MHW+5 m | present | 263494 | 208126 (79.0) | 44777 (17.0) | 8047 (3.1) | 331 (0.13) |





**Table 4.** Affected roads [km] in Norway under different water levels at present and for 2090. Numbers in parentheses indicate the percentage share of the total for each subcategory. Note that the lengths of affected roads for present MHW have been subtracted from the numbers for higher water levels.

| Scenario | Year | Total | Private | Public |
|---|---|---|---|---|
| MHW | 2090 | 36.0 | 24.4 (67.8) | 11.5 (31.9) |
| 20 yr | present | 297.7 | 203.7 (68.4) | 93.9 (31.5) |
| 20 yr | 2090 | 999.8 | 521.9 (52.2) | 477.8 (47.8) |
| 200 yr | present | 511.1 | 318.6 (62.3) | 192.3 (37.6) |
| 200 yr | 2090 | 1340.8 | 653.6 (48.7) | 687.0 (51.2) |
| 1000 yr | present | 670.3 | 393.0 (58.6) | 277.2 (41.4) |
| 1000 yr | 2090 | 1569.0 | 740.5 (47.2) | 828.4 (52.8) |
| 1000 yr+1 m | present | 2506.0 | 1060.8 (42.3) | 1445.1 (57.7) |
| 1000 yr+1 m | 2090 | 3490.9 | 1382.2 (39.6) | 2108.6 (60.4) |
| MHW+1 m | present | 215.4 | 148.4 (68.9) | 66.9 (31.1) |
| MHW+2 m | present | 1582.6 | 742.2 (46.9) | 840.3 (53.1) |
| MHW+3 m | present | 3436.9 | 1358.5 (39.5) | 2078.4 (60.5) |
| MHW+4 m | present | 5172.7 | 1901.5 (36.8) | 3271.0 (63.2) |
| MHW+5 m | present | 6832.2 | 2433.3 (35.6) | 4398.8 (64.4) |



**Table 5.** The percentage increase in exposed areas, buildings, and roads between 2017 and 2090 for different sea-level scenarios.

| Category | Sub category | 20 yr | 200 yr | 1000 yr | 1000 yr+1 m |
|---|---|---|---|---|---|
| Area | Total | 61 | 52 | 48 | 24 |
| | Developed | 249 | 202 | 178 | 33 |
| | Nature | 54 | 47 | 43 | 22 |
| | Public facility | 233 | 100 | 100 | 33 |
| | Primary industry | 97 | 71 | 66 | 33 |
| Buildings | Total | 35 | 31 | 28 | 14 |
| | Private | 29 | 26 | 24 | 14 |
| | Private industry | 57 | 48 | 43 | 13 |
| | Public | 116 | 98 | 91 | 21 |
| | Critical Infrastructure | 195 | 167 | 154 | 26 |
| Roads | Total | 236 | 162 | 134 | 39 |
| | Private | 156 | 105 | 88 | 30 |
| | Public | 409 | 257 | 199 | 46 |





**Table 6.** The percentage increase in exposed areas, buildings, and roads between different storm-surge return heights at present and for 2090.

| Category | 20→200 yr present | 200→1000 yr present | 20→200 yr 2090 | 200→1000 yr 2090 |
|---|---|---|---|---|
| Area | 22 | 11 | 15 | 8 |
| Buildings | 12 | 7 | 9 | 5 |
| Roads | 72 | 31 | 34 | 17 |





**Table 7.** Quantitative assessment of effects contributing to the accuracy of the mapping.

| Contributor to uncertainty | Uncertainty [m] | Comment/Reference |
|---|---|---|
| DEM | RMSE<0.1 | Project goal |
| HREF | 0.01-0.10 | Personal communication Olav Vestøl at the Norwegian Mapping Authority |
| Height of MSL in NN2000 | 0.02-0.10 | Simpson et al. (2015) |
| Height of MHW in NN2000 | Unknown | |
| Mean range of 95% confidence intervals for 20-year storm surges along the Norwegian coast | 0.15 | Simpson et al. (2015) |
| Mean range of 95% confidence intervals for 200-year storm surges along the Norwegian coast | 0.21 | Simpson et al. (2015) |
| Mean range of 95% confidence intervals for 1000-year storm surges along the Norwegian coast | 0.25 | Simpson et al. (2015) |
| Projections of future sea level for 2090 | >0.5 | Range of models, assessed to be 66% of the total possible outcome for the pathway |
| Horizontal position of buildings | 0.2-2 | Effect depends on slope of terrain |
| Horizontal position of roads and areas | 2-50 | Effect depends on slope of terrain |





**Table 8.** Comparisons of heights from DEMs and heights observed by GNSS. Two DEMs have been assessed, i.e., a nationwide DEM with a spatial resolution of 1.0×1.0 m and a regional DEM with a spatial resolution of 0.5×0.5 m covering a smaller test field.

| Data set | Surveying method | Minimum difference [m] | Maximum difference [m] | Mean difference [m] | RMSE [m] | Number of observations |
|---|---|---|---|---|---|---|
| $h_{\mathrm{NGN}} - h_{1.0 \times 1.0}$ (overall) | Network | -61.772 | 4.866 | -0.338 | 1.92 | 10301 |
| $h_{\mathrm{NGN}} - h_{1.0 \times 1.0}$ | Network | -1.000 | 1.000 | -0.116 | 0.259 | 9703 |
| $h_{\mathrm{CPOS}} - h_{1.0 \times 1.0}$ (overall) | CPOS | -0.577 | 1.104 | 0.108 | 0.282 | 132 |
| $h_{\mathrm{CPOS}} - h_{1.0 \times 1.0}$ (flat terrain) | CPOS | -0.255 | 0.395 | -0.008 | 0.096 | 75 |
| $h_{\mathrm{CPOS}} - h_{0.5 \times 0.5}$ (overall) | CPOS | -0.21 | 0.884 | 0.011 | 0.158 | 134 |
| $h_{\mathrm{CPOS}} - h_{0.5 \times 0.5}$ (flat terrain) | CPOS | -0.21 | 0.349 | -0.031 | 0.088 | 73 |

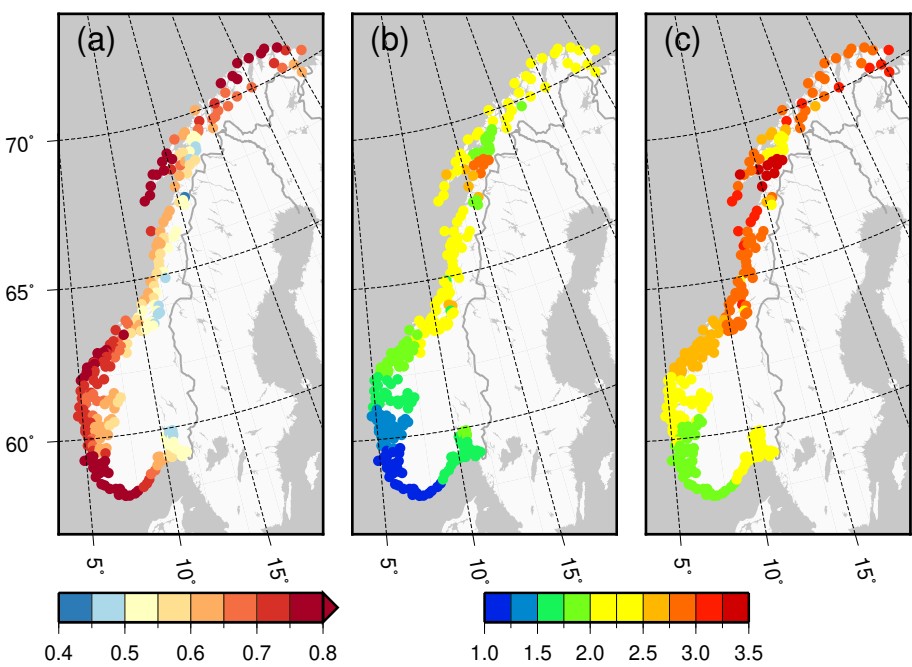

**Figure 1.** Projected RSL change for the period 2081-2100 relative to 1986-2005 (a), 200-year storm-surge return height above MHW (b), and the sum of these two (c).





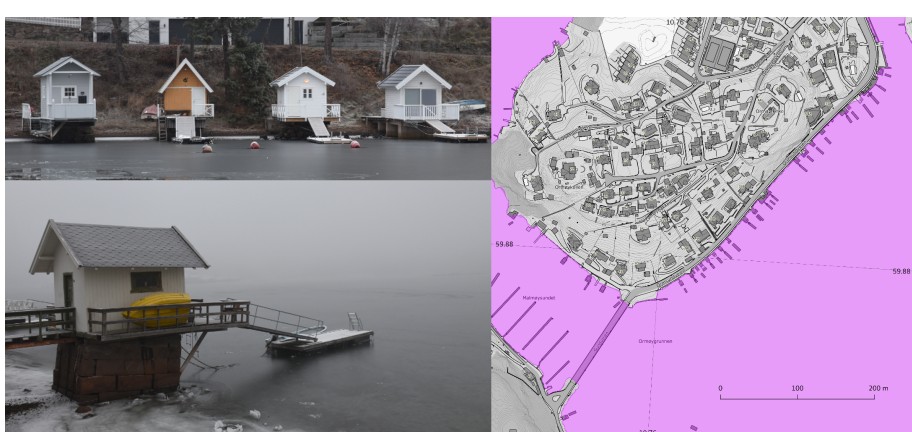

**Figure 2.** Buildings on piers or pillars above the water surface, like here at Ormøya close to Oslo, are erroneously mapped as flooded for present MHW (violet). Photo: Kristian Breili.

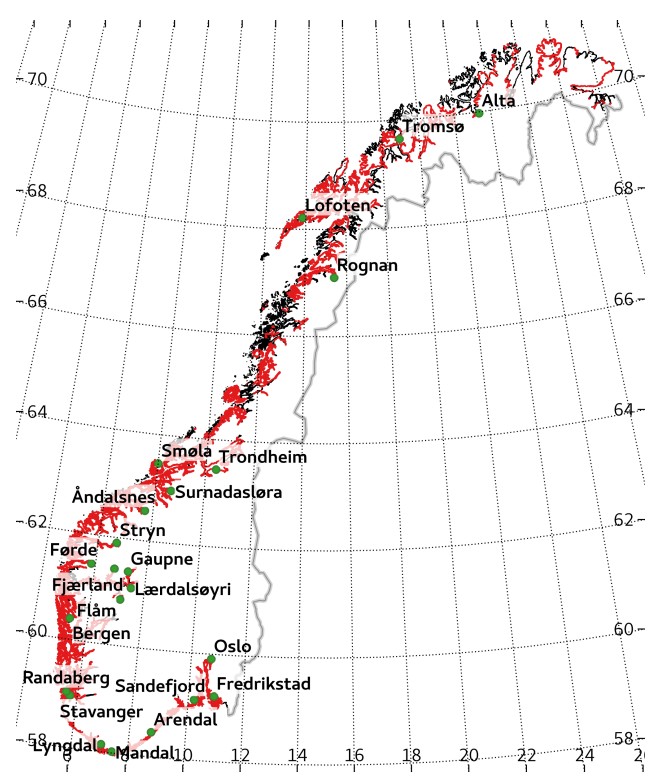

**Figure 3.** Red lines indicate areas covered by the inundation maps per December 2018. Information for all water levels is not available for all mapped zones due to a lack of knowledge on ocean tides for parts of the coast. In these zones, only the storm-surge return heights can be calculated because they are not referenced to MHW. The green markers indicate locations discussed in the text.

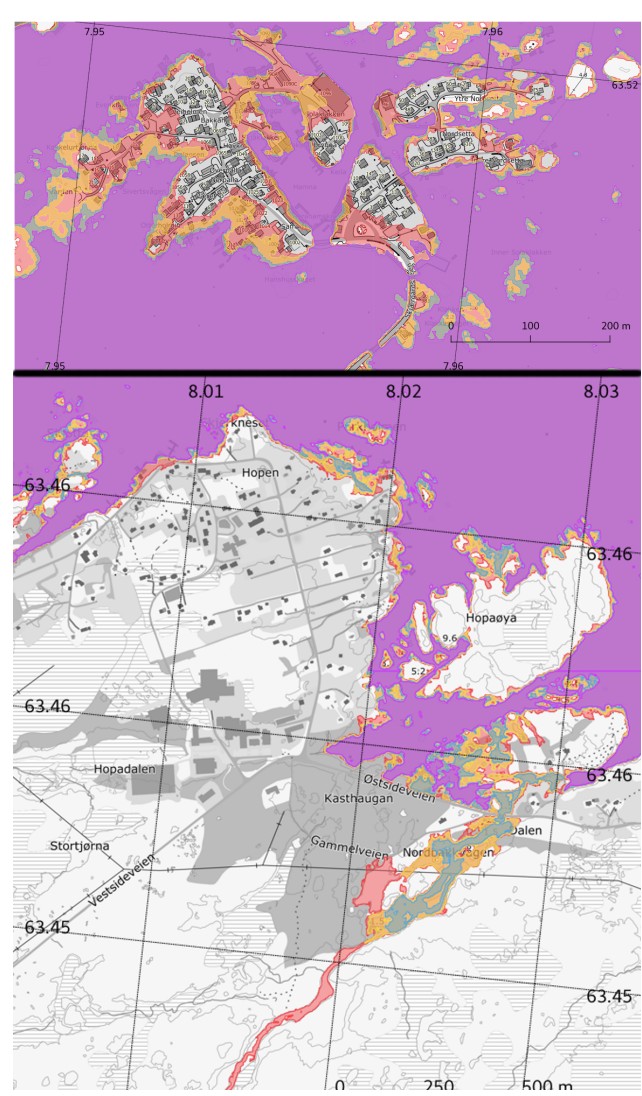

**Figure 4.** Stack of inundation maps covering the northern part of the island Smøla (lower panel) and the fishing village of Veiholmen (upper panel) located on the strandflat in the middle of Norway. Violet: Present MHW. Green: MHW for 2090 (0.74 m above present MHW). Orange: Present 200-year storm surge (1.29 m above present MHW). Red: 200-year storm surge for 2090 (2.03 m above present MHW).



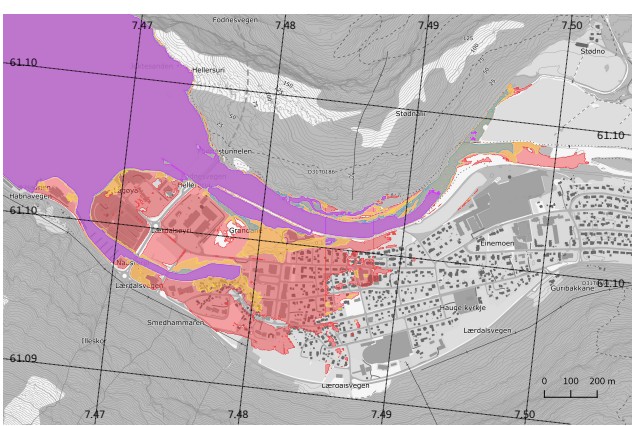

**Figure 5.** Stack of inundation maps covering the village Lærdalsøyri located on a glaciofluvial delta at the head of Sognefjorden. Violet: Present MHW. Green: MHW for 2090 (0.58 m above present MHW). Orange: Present 200-year storm surge (1.00 m above present MHW). Red: 200-year storm surge for 2090 (1.58 m above present MHW)


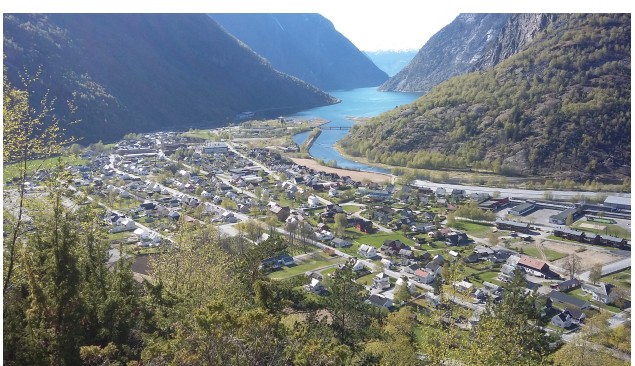

**Figure 6.** The village Lærdalsøyri. Photo: Magnhild Aspevik.

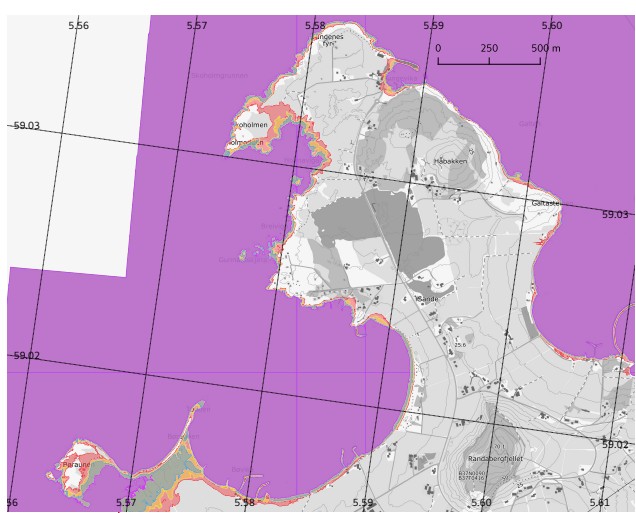

**Figure 7.** Stack of inundation maps covering Randaberg located on soft moraine in the southwest of Norway. Violet: Present MHW. Green: MHW for 2090 (0.79 m above present MHW). Orange: present 200-year storm surge (0.99 m above present MHW). Red: 200-year storm surge for 2090 (1.78 m above present MHW).



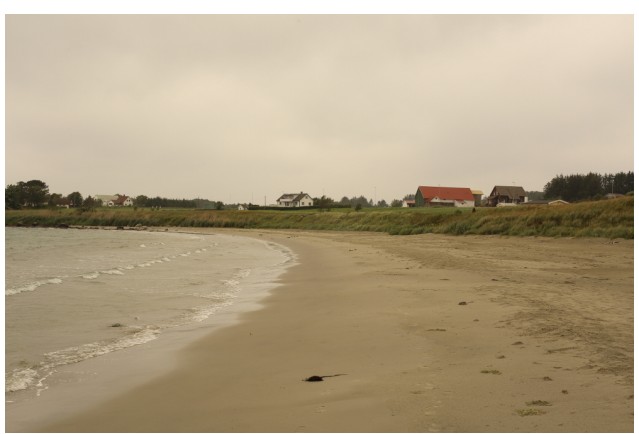

**Figure 8.** Soft sand dunes at Sandestranda close to Randaberg. Photo: Oda R. Ravndal.





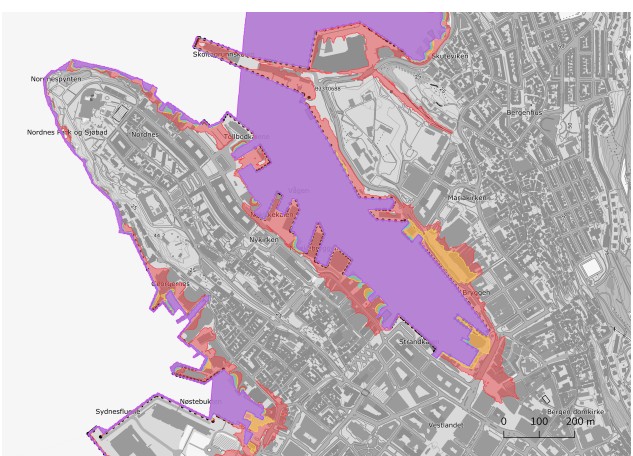

**Figure 9.** Stack of inundation maps indicating areas affected by coastal flooding in Bergen. Violet: Present MHW. Green: MHW at 2090 (0.71 m above present MHW). Orange: present 200-year storm surge (0.96 m above present MHW). Red: 200-year storm surge for 2090 (1.68 m above present MHW).


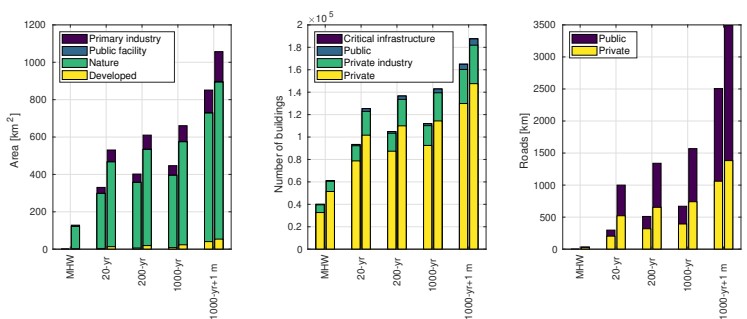

**Figure 10.** The bars indicate the size of areas (left), the number of buildings (middle), and the length of roads affected by sea-level rise and storm surge in Norway. For each water level, the left and right bars indicate affected objects at present and for 2090, respectively.

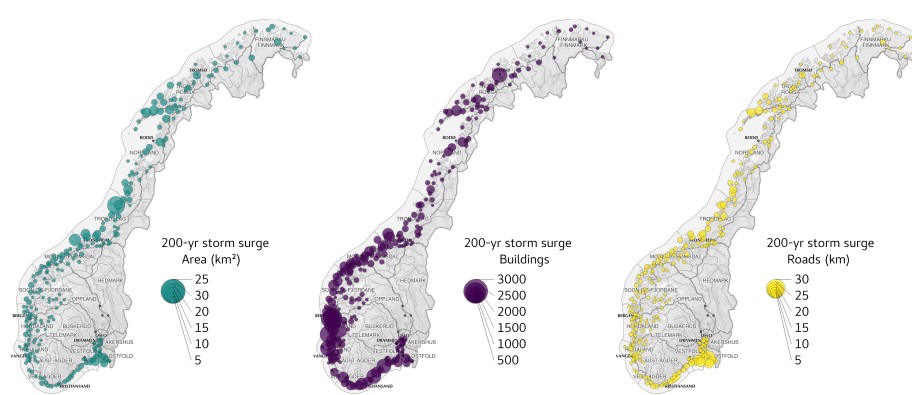

**Figure 11.** Affected land areas (left), buildings (middle), and roads (right) due to a 200 year storm-surge hazard at present. The radius of the bubbles are proportional to the size of flooded land areas, the number of exposed buildings, and length of roads.



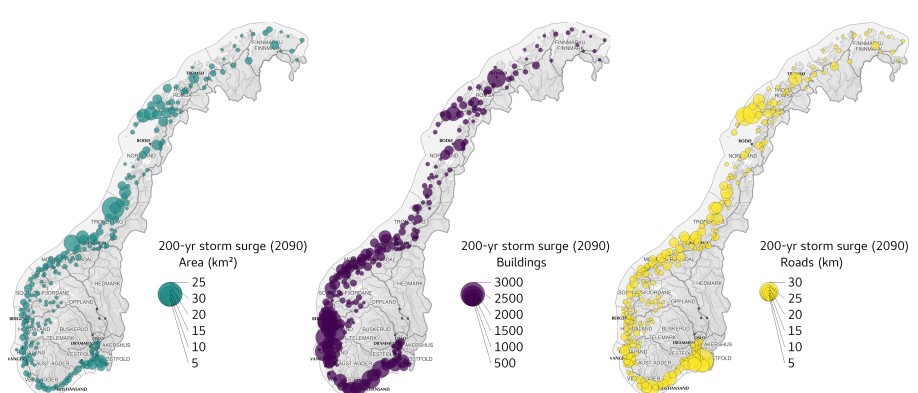

**Figure 12.** Similar to Figure 11, but for 2090.



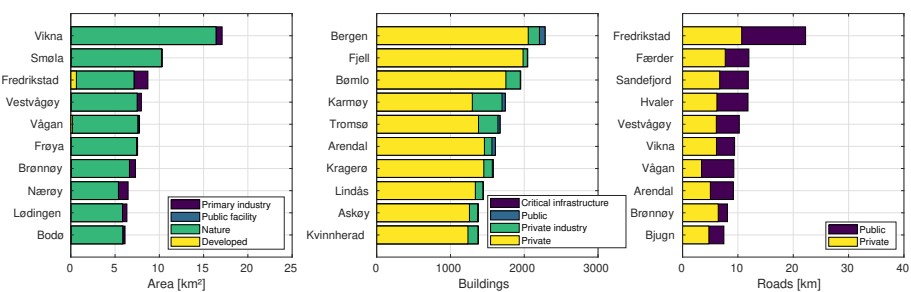

**Figure 13.** The ten municipalities with most land-areas, buildings, and roads affected by a 200-year storm-surge hazard at present sea level.





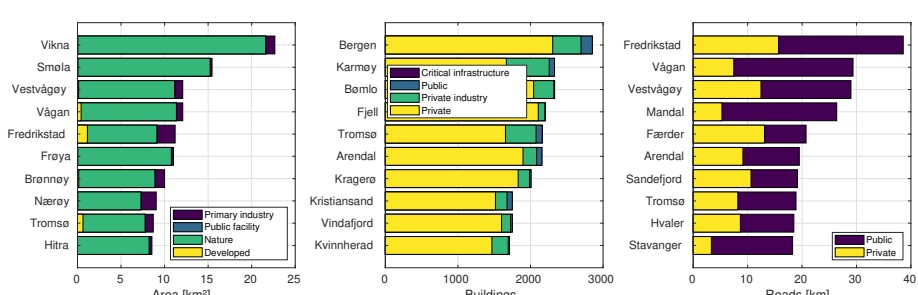

**Figure 14.** Similar as Figure 13, but for 2090.