# Peer review of "High accuracy coastal flood mapping for Norway using LiDAR data"

_Natural Hazards and Earth System Sciences, 2019_

## Referee Comment (RC1) · Jordan Eamer (Referee) · 23 Sep 2019

Reviewer – Jordan Eamer, Geological Survey of Canada

General comments

This is a generally well-written, informative description of a new dataset and suite of tools for coastal management activities for the country of Norway. Thank you for affording me the opportunity to review! The analysis is thorough and description of input data, results, and related uncertainties is sufficient. I, however, would prefer to see this presented as a brief communication, as I believe the major contribution of this work lies more in its presentation of the dataset and availability of management tools and less so in any analyses of coastal risk, adaptations, or impacts. For example, I feel like the

conclusions section (L450 – 473) can mostly represent the results section without any significant loss of interpretative value, and the remaining lines (to L493) appropriately represents sufficient discussion of uncertainties and overarching themes.

This opinion is not strong enough, however, to merit a major revision or rejection. I believe it's still an important contribution. Additionally, the assigned editor has looked over the manuscript and deemed it appropriate as a research article, so I will defer to their judgement here.

Other general notes: - For the uncertainty analyses:

o With confidence intervals (or upper/lower limits of uncertainty) expressed for 9 of 10 metrics in Table 7, I'm not sure why there was no utilization of these in the results. At the least, one figure could have shown the difference in inundation extent using an upper and lower limit, and at most, all results could have been expressed as their appropriate ranges incorporating all relevant uncertainties

o One of the greatest sources of uncertainty as discussed was the bias introduced by engineered structures over water. I'm curious as to why the authors did not attempt to crop out these structures using a coastline mask. Was it because of discussed issues with the coastline not agreeing between datasets?

o A DEM accuracy of <0.1m (Table 7) is not really true, is it? That refers to the accuracy of the processed lidar point data, and not the interpolated DEM. As stated in the text, an accuracy of 0.26m is more appropriate. So why is it presented as such in the table, and elsewhere in the text?

- For the figures: o I would like to have seen combination figures – pictures inset, side-by-side, or multi-panel with the representative inundation maps (e.g. 5 & 6, 7 & 8). Larger, too.

o Clear graphs showing the results from tables 2-6 more clearly would increase the impact of these findings. The bar graphs in Figures 10, 13, 14 are informative, but I

can't help but feel like more data could be incorporated into larger line graphs for more interpretive power, and showcase the infrastructure challenges facing Norway in the RCP8.5 scenario.

o Figures 11/12 are hard to interpret, too much overlapping data. Perhaps colour-magnitude hexagons might more clearly convey the spatial patterns (e.g. see Figure 2, https://www.nature.com/articles/s41467-019-10762-4)

- References are minimal, and several are non-peer reviewed reports. A more thorough examination of the literature, particularly with regards to inundation mapping and DEM analyses/uncertainties, would really benefit this manuscript

Line by line comments as follows: 13 – Adaption and adaptation are used interchange-ably throughout the manuscript. Please pick one for readability.

19 – The sentence beginning "The consequences . . ." is awkward. Maybe remove "and many" as well as the "the" before coastal.

24 – is GIA the only component of VLM at play? No tectonics? I ask because I don't know and a cursory look turned up no information. Just curious!

78 – See point above, how this manuscript may be better suited as a short commu-nication, describing the tool and its applications, inviting the reader to go and perform their own analyses.

128 – I recognize that it's an even more complicated variable than those already left out from your analyses, but I would have appreciated some mention of a reduction in sea ice and associated coastal effects on arctic communities. In Canada the highest rates of coastal retreat and impact on communities/infrastructure due to SLR is in arctic areas impacted by a loss of sea ice and associated increased wave activity. . . I'm sure there are some similar effects being witnessed in northern parts of the study area.

161 – is that a definitive statement – every grid cell has at least 2 datapoints? Or is that an average. Also, interpolation method?

168 – Some discussion of alternatives to the bathtub methods (e.g. modeling approaches using XBeach)?

193 – Again, unless this is a short communication introducing the tool, this kind of discussion isn't really necessary in a scientific paper

202 – See general comments for sections 3 and 4. Generally I think that sections 3.1 and 3.2 could be thinned significantly, particularly if more detailed and interpretive figures and maps are presented. 3.3 is a good section – this is the level of interpretation I'd personally like to see in the results. Sections 4.1 and 4.2 are great – but then none of these uncertainties so carefully outlined are included in the analysis!

421 – " . . . and future applications of this tool" or something to that effect?

---

## Referee Comment (RC2) · Dean Gesch (Referee) · 3 Oct 2019

This paper is an effective study of sea-level rise and coastal flooding exposure in Norway based on a high-quality DEM. It exhibits a sound, straightforward approach that uses many of the best practices that have been established for these types of coastal assessments. The paper documents well the data, methods used, and results, and the tables and figures effectively support the material in the text. Overall, the Discussion section is very good, especially the factors affecting uncertainty and the accuracy of the DEM.

The results could be improved by attaching confidence levels to the estimates of impacted area and objects. This would entail not just describing the accuracy of the DEM

(and the associated datum conversions), but applying that cumulative vertical uncertainty to characterize the confidence of the results (see Gesch, 2013 and Gesch, 2018 for details on how this is done). All the needed information is already available with the comprehensive DEM accuracy assessment that has been done and all the other uncertainty information that is listed in Table 7. I am not saying that this needs to be done for this paper to be accepted, as I believe the results as currently presented are useful, but adding confidence information could be done in future related work (perhaps as the remaining 20% of the country is worked on and national results are revised and added to), and the authors could add this idea of characterizing the confidence of the results to the Discussion/Conclusions sections.

Comment on lines 207-216 (discussion of Smola) in section 3.1, and Figures 13 and 14 that it refers to: The area affected is important, but without knowing the total area of each of the ten municipalities (assuming there's variability in the areas) it's hard to see which ones are affected the most. So you could also show the affected area as a percent of the total area of each municipality as a way to rank the municipalities.

Please see the attached annotated manuscript for a few other comments tied to specific locations in the text. Some are thoughts to consider for possible inclusion, while others are suggested edits or corrections that should be made.

Please also note the supplement to this comment:
https://www.nat-hazards-earth-syst-sci-discuss.net/nhess-2019-217/nhess-2019-217-RC2-supplement.pdf
* * *
[Figure]

**Supplement:**

[revised manuscript text omitted]

---

## Author Response (AR1)

[revised manuscript text omitted]
. Two methods are applied to interpolate the LiDAR data to a regular DEM. In a first try, natural neighbor interpolation (Sibson, 1981) was used. If this failed, empty spaces were binned with an average value. The vertical accuracy of the  LiDAR data has a production goal  root mean square error (RMSE) of 0.1 m for well-defined solid areas  (Kartverket, 2014). The DEM was transformed from ellipsoidal heights to NN2000 by using the height reference surface HREF (Solheim, 2000).

Presently, about 80% of Norway is covered by the DEM, and the entire country is expected to be mapped by 2023. We assume no geomorphologic changes (e.g. erosion) or man-made landscape interventions take place over time, i.e., the same elevation data are used to map sea-level rise and storm surge today and for 2090.

In order to identify flooded zones, we have followed a "bathtub" approach similar to the one outlined in, e.g., Gesch (2009), Rowley et al. (2007), and Poulter and Halpin (2008). The "bathtub" approach is favored for several reasons. Firstly, mapping results from this approach are consistent with how current guidelines on coastal planning are applied in Norway. Secondly, the approach is straightforward, computationally inexpensive, and has been widely used in large-scale coastal flooding analyses. However, there are known limitations of the "bathtub" method. For example, the response of hydrodynamics, morphology, and ecology as sea level rises is not accounted for (see Passeri et al. (2015) for a review). Some of these effects could be important on local scales along the Norwegian coast.

[revised manuscript text omitted]
355 we find that six of the ten municipalities with the highest percentages are located on the west coast. For 2090, the size of the flooded area increases and the order of the municipalities changes slightly, but the general pattern of regional impacts does not change.

The regional pattern of land areas affected by coastal flooding closely corresponds to regional differences in the storm-surge return heights, although regional differences in topography and projected sea-level rise also plays a role to some extent. The exposure of land areas to coastal flooding is one measure of the impacts of 21st century sea-level rise and storm surge. As mentioned above, the majority (> 80%) of these land areas are classified as nature. Several of the municipalities in the middle of Norway, where the largest land areas are flooded, are sparsely populated. This is also evident from the maps visualizing the distribution of affected buildings (middle part of Figure ?? and ??Figure 10), which have quite different spatial patterns. For buildings, the consequences of storm-surge flooding is particularly large in two counties, Hordaland and Rogaland, which are on the west coast of Norway. Moreover, many buildings are exposed along the outer parts of Oslofjorden, along the southern coast, around Trondheimsfjorden, in Lofoten, and in Tromsø. These regions stand out as they are densely populated and include several of the largest cities in Norway.

The pattern of exposed roads (right panel of Figure ?? and ??Figure 11) is similar to that for land areas, but the ten most exposed municipalities also includes include some locations along the southernmost part of the coast.

**4    Discussion**

**4.1    Uncertainties Accuracy of mappingthe DEM**

A number of different factors determine the accuracy of the inundation maps and associated statistics of exposed objects. Although the uncertainties attached to these factors are not accounted for in our analysis, we discuss their relative importance to the results. Factors determining the accuracy of our results include uncertainties related to (1) the DEM, (2) the vertical reference frame NN2000, (3) the transformation of ellipsoidal heights to the national height system (HREF), (4) the height determined for mean sea level and MHW, (5) the estimated storm-surge return heights, (6) the sea-level projections, (7) the horizontal position of buildings and roads, (8) inaccurate polygons defining land cover, and (9) the effect of, e.g., buildings on pillars and piers. We note that these factors and their uncertainties are inherently different. Furthermore, not all of these factors are relevant for all of the water levels we have mapped. Uncertainties related to storm-surge heights are, for example, not relevant when mapping MHW.

When assessing future flood risk the largest uncertainty probably relates to the sea-level projections (see Table 7) . The sea-level projections have uncertainties related to the future emission scenario and the ability of models to simulate the future sea-level response. For the mapping method approach taken here, however, where sea-level rise is considered a fixed number (95th percentile of RCP8.5), the uncertainty associated with the sea-level projections can be ignored. In this situation, planning policy dictates which sea level number to use, but there will nevertheless be mapping uncertainties related to, e.g., the accuracy of the DEM and tidal datums.

The DEM has a project goal root mean square error (RMSE) of less than The project goal uncertainty (RMSE) of the LiDAR data, from which the DEM is interpolated, is 0.1 m (Kartverket, 2014). This is ensured by comparing and fitting the point cloud of LiDAR measurements to control-fields and road tracks with heights observed by Global Navigation Satellite Systems (GNSS). Both control-fields and road tracks must be considered as favorable LiDAR targets. The actual accuracy of

the  DEM depends on the slope of the terrain, terrain surface complexity, target reflectivity, canopy coverage and near ground vegetation, the density and distribution of the ground returns, the accuracy of the LiDAR system, the interpolation algorithm used to create the DEM from the source data, and the spatial resolution of the DEM (e.g., Reutebuch et al., 2003; Li, 1992). Furthermore, transforming ellipsoidal heights to the national height system NN2000 may introduce additional errors.

395 As heights observed by both GNSS and LiDAR are transformed to NN2000 using the same HREF model, any errors in the transformation will not be detected by comparison to the GNSS control measurements. We therefore consider the project goal  uncertainty of the LiDAR data as an optimistic error estimate for the  DEM in the coastal zone.

400 ~~difference between MHW and NN2000. In addition, other tidal constituents give small contributions to the mean high tide that the present definition of MHW does not include. Unfortunately, there are no assessments of the uncertainty of MHW along the Norwegian coast. But what we can say is that the tidal datums, storm-surge levels, and their heights with respect to NN2000 are well known in areas close to the tide gauges. Along other parts of the coast, they are less well defined. Uncertainties associated with the tidal datums and storm-surge levels may therefore exceed the project goal uncertainty (RMSE<0.1 m) of~~
405

~~There are also effects that are not included in our analysis; for example, wave setup and runup, changes in tides due to sea-level rise, coastal erosion, and the effects of river flooding close to the coast. We have assumed no future changes to the storm-surge return heights but note that a recent study projects areas of increase, areas of decrease, and also areas of model disagreement along the Norwegian coast (Vousdoukas et al., 2018b).~~

410 **4.2 **

[revised manuscript text omitted]

**4.2  Uncertainties of mapping**

A number of different factors determine the accuracy of the inundation maps and associated statistics of exposed objects. Although the uncertainties attached to these factors are not accounted for in our analysis, we discuss their relative importance
465  to the results. Factors determining the accuracy of our results include uncertainties related to (1) the DEM, (2) the vertical reference frame NN2000, (3) the transformation of ellipsoidal heights to the national height system (HREF), (4) the height determined for mean sea level and MHW, (5) the estimated storm-surge return heights, (6) the sea-level projections, (7) the horizontal position of buildings and roads, (8) inaccurate polygons defining land cover, and (9) the effect of, e.g., buildings on pillars and piers. We note that these factors and their uncertainties are inherently different. Furthermore, not all of these
470  factors are relevant for all of the water levels we have mapped. Uncertainties related to storm-surge heights are, for example, not relevant when mapping MHW.

When assessing future flood risk the largest uncertainty probably relates to the sea-level projections (see Table 7). The sea-level projections have uncertainties related to the future emission scenario and the ability of models to simulate the future sea-level response. For the mapping method approach taken here, however, where sea-level rise is considered a fixed number
475  (95th percentile of RCP8.5), the uncertainty associated with the sea-level projections can be ignored. In this situation, planning policy dictates which sea level number to use, but there will nevertheless be mapping uncertainties related to, e.g., the accuracy of the DEM and tidal datums. Given the accuracy of the DEM used in this study (0.26 m RMSE), each water level will be mapped with a different level of confidence because the lower levels are close to the inherent noise level of the DEM.

In Norway, MHW corresponds to the height of the M2 tidal constituent above mean sea level. The uncertainty of MHW
480  therefore depends on the definition of mean sea level, the uncertainty of the estimated amplitude of M2, and the height difference between MHW and NN2000. In addition, other tidal constituents give small contributions to the mean high tide that the present definition of MHW does not include. Unfortunately, there are no assessments of the uncertainty of MHW along the Norwegian coast. But what we can say is that the tidal datums, storm-surge levels, and their heights with respect to NN2000 are well known in areas close to the tide gauges. Along other parts of the coast, they are less well defined. Uncertainties
485  associated with the tidal datums and storm-surge levels may therefore exceed the project goal uncertainty (RMSE<0.1 m) of the elevation data in some areas.

There are also effects that are not included in our analysis; for example, wave setup and runup, changes in tides due to sea-level rise, coastal erosion, and the effects of river flooding close to the coast. We have assumed no future changes to the storm-surge return heights but note that a recent study projects areas of increase, areas of decrease, and also areas of model
490  disagreement along the Norwegian coast (Vousdoukas et al., 2018b).

In summary, a preliminary assessment indicates that the elevation model (RMSE 0.26 m) is the largest source of uncertainty in our mapping method. There are also smaller errors associated with different vertical datums and transformations between datums that have not been assessed for the entire coast. However, we believe that the sum of these mapping errors are generally

smaller than the projected sea-level rise, which gives us confidence in our results. Future work should look at how these
uncertainties can be incorporated into our mapping and web tool (Gesch, 2013, 2018; Cooper and Chen, 2013; Cooper et al., 2013).

**4.3 Comparison to other studies and future work**

As an alternative to our approach where affected objects were identified by overlaying inundation polygons with geospatial data like buildings, the height of the objects themselves can be used to identify what is exposed to future sea-level rise and storm surges. Almås and Hygen (2012) followed this approach and used a DEM (unknown spatial resolution but likely $10 \times 10$ m horizontal resolution with at best 2-3 m vertical error) to determine heights of buildings in the coastal zone. In their study, approximately 110,000 buildings were found nationwide with a height less than one meter above elevation zero in the former national vertical reference system of Norway, NN1954, which at Norwegian tide gauges has its zero height within -0.09 m and 0.17 m from mean sea level. Unfortunately, a straightforward comparison of the findings of Almås and Hygen (2012) with our results (Table 3) is not possible. Firstly, this is because we have not analyzed affected buildings for a fixed height, but have taken into account tidal variations. This will likely make a significant difference because MHW ranges from a couple of centimeters to 1.1 m above mean sea level in Norway. If not taken into account, the flooding risk will be underestimated in areas with mean high tide elevation exceeding 0 m, and comparisons across regions with different tidal levels will be compromised (Strauss et al., 2012). Secondly, we have used NN2000 as vertical reference frame instead of NN1954. At the tide gauges along the Norwegian coast, the difference between these two vertical reference frames varies between -15 cm and 12 cm (Simpson et al., 2015). Thirdly, the numbers in Table 3 are based on data that cover 80% of the coast, while the study by Almås and Hygen (2012) covers the entire coast. If we still attempt to compare numbers, the water level MHW + 1 m is perhaps the most similar to the height used in their analysis. For MHW + 1 m our results show 86,944 affected buildings, which is significantly less than the ∼110,000 reported by Almås and Hygen (2012). Note that MHW+1 m in most areas will be higher than height 1 m in NN1954.

The  "bathtub" approach applied in the present study results in maps that are consistent with national guidelines on how to account for future sea-level change and storm-surge in coastal planning. Currently, there are no regulations for modelling the effects of waves, which may increase mean sea level during a storm and introduce geomorphological changes due to erosion and transport of sediments. Modelling the effects of waves should be addressed in future work and will require a more advanced framework than provided by the "bathtub" approach. A more advanced framework is provided by the open-source numerical model XBeach (Roelvink et al., 2009), which is developed to simulate the effects of storms on sandy coasts with domain size of kilometers. The XBeach model is not a tool for analyzing the entire Norwegian coast, but is suitable for case studies of vulnerable areas like beaches and coasts covered by soft sediments (e.g., southwest of Norway).

The present study does not aim at being a socioeconomic analysis of coastal flooding for Norway as the climate service includes no information on value of property or the population in the coastal zone. Our inundation maps, however, could be used as input to a socioeconomically analysis. In their analysis, Vousdoukas et al. (2018a) caution that the accuracy of their modeled extreme sea levels for Norway may be affected by the presence of many bays, islands and steep complex terrain.

Furthermore, they indicate that elevation data of higher spatial resolution are required to achieve the same accuracy for Norway as for flatter parts of Europe. This suggests that high accuracy national coastal flooding maps must be used to achieve results that are useful for planners and stakeholders. We believe that the methods and data used for mapping sea levels in the present study, especially the use of a $1\times1$ m DEM and accounting for regional differences in MHW, storm-surge heights and sea-level rise, represent significant progress compared to the methods used by Almås and Hygen (2012) and Vousdoukas et al. (2018a).

**5 Conclusions**

Using new high accuracy LiDAR elevation data we have generated coastal flooding maps for Norway. Thus far, we have mapped $\sim$80% of the coast, for which we currently have data of sufficient accuracy to perform our analysis. Our mapping method accounts for regional variations in tidal datums, storm-surge levels, and projections of sea-level rise. Nationwide we have identified a total area of 400 km$^2$, 105,000 buildings, and 510 km of roads that are at risk of flooding from a 200 year storm-surge event at present. These numbers will increase to 610 km$^2$, 137,000, and 1340 km with projected sea-level rise to 2090 (95th percentile of RCP8.5 as recommended in planning). If sea-level rise exceeds the projections by 1 m, then an area of 1060 km$^2$, 189,000 buildings, and 3490 km of roads would be exposed to 1000 year storm surge. This gives an indication of how vulnerable Norway is to a scenario of rapid ice melt from Antarctica. Notably, we also find that the numbers of affected objects for a 20 year storm-surge return height in 2090 will exceed the numbers for the 1000 year storm-surge at present. Indicating that an increasing number of objects will be at risk of more frequent flooding.

Examining the categories of what is at flooding risk shows the vast majority of areas are classified as nature. However, the fraction of total area classified as developed, public facility, or primary industry increases for higher water levels. Developed areas at flooding risk from a 200 year storm surge will increase more than three times in size between now and 2090 due to sea-level rise (increasing from 6 to 19 km$^2$). For buildings, around 80% of those at risk are private (homes, cabins, garages, or boat houses) for all mapped water levels. The fraction of buildings classified as private industry, public, or critical infrastructure increases for higher water levels. Critical infrastructure buildings at risk from a 200 year storm surge will increase from 30 to 80 between now and 2090. For roads, the percentage of public roads at risk will increase for higher water levels. Thus, while sea-level rise leads to more objects to be at risk of flooding, our results also indicate an increasing fraction will be objects of higher value.

Regional differences indicate that the west and southern coast of Norway, outer parts of Oslofjorden, areas around Trondheimsfjorden, and Tromsø have the largest numbers of buildings at risk of coastal flooding. For land areas and roads, it is the middle of Norway and outer Oslofjorden that are most at risk. Regional differences in the number of objects exposed to flooding can largely be explained by regional differences in population density. Inspection of the inundation maps shows that, across much of Norway, the typically steep topography restricts flooding to areas immediately adjacent to the coast. Of the examples we have examined, we find cities, island communities, and in particular towns and villages located on glaciomarine deltas are at risk from coastal flooding. The flooding risk at glaciomarine deltas can be exacerbated by the effect of river flooding.

[revised manuscript text omitted]

Geonorge: Shared map catalogue of Norway, Web portal, retrieved from https://kartkatalog.geonorge.no/metadata/geovekst/felles-kartdatabase-fkb/0e90ca71-6a02-4036-bd94-f219fe64645f, 2019.

Gesch, D. B.: Analysis of Lidar Elevation Data for Improved Identification and Delineation of Lands Vulnerable to Sea-Level Rise, J. Coastal Res., 53, 49–58, https://doi.org/10.2112/SI53-006.1, 2009.

Gesch, D. B.: Consideration of Vertical Uncertainty in Elevation-Based Sea-Level Rise Assessments: Mobile Bay, Alabama Case Study, J. Coastal Res., 63, 197–210, https://doi.org/10.2112/SI63-016.1, 2013.

Gesch, D. B.: Best Practice for Elevation-Based Assessment of Sea-Level Rise and Coastal Flooding Exposure, Front. Earth. Sci., 6, 230, https://doi.org/10.3389/feart.2018.00230, 2018.

IPCC: Summary for Policymakers, in: Climate Change 2013: The Physical Science Basis. Contribution of Working Group I to the Fifth Assessment Report of the Intergovernmental Panel on Climate Change, edited by Stocker, T. F., Qin, D., Plattner, G. K., Tignor, M., Allen, S. K., Boschung, J., Nauels, A., Xia, Y., Bex, V., and Midgley, P. M., Cambridge University Press, ISBN 978-1-107-05799-1, 2013.

Kartverket: Forprosjekt "Nasjonal, detaljert høydemodell" (in Norwegian), Tech. rep., Norwegian Mapping Authority, web portal: www.hoidedata.no, 2014.

Kartverket: The Norwegian coastline (in Norwegian), Web portal, retrieved from https://www.kartverket.no/kunnskap/Fakta-om-Norge/norges-kystlinje/kystlinjen-i-kilometer, 2019a.

Kartverket: The SOSI-standard (in Norwegian), Web portal, retrieved from https://www.kartverket.no/geodataarbeid/Standarder/SOSI/SOSI-standarden-del-2, 2019b.

Kartverket: Se havnivå i kart (View sea-level rise in maps), Web portal, retrieved from https://www.kartverket.no/en/sehavniva, 2019c.

Kierulf, H. P., Steffen, H., Simpson, M. J. R., Lidberg, M., Wu, P., and H., W.: A GPS velocity field for Fennoscandia and a consistent comparison to glacial isostatic adjustment models, J. Geophys. Res. Solid Earth, 119, 6613–6629, https://doi.org/10.1002/2013JB010889, 2014.

Le Cozannet, G., Nicholls, R. J., Hinkel, J., Sweet, W. V., McInnes, K. L., Van de Wal, R. S. W., Slangen, A. B. A., Lowe, J. A., and White, K. D.: Sea Level Change and Coastal Climate Services: The Way Forward, J. Mar. Sci. Eng., 5(49), https://doi.org/10.3390/jmse5040049, 2017.

Li, Z.: Variation of the accuracy of digital terrain models with sampling interval, Photogramm. Rec., 14(79), 113–128, 1992.

Næss, A. and Gaidai, O.: Estimation of extreme values from sampled time series, Struct. Saf., 31, 325–334, https://doi.org/10.1016/j.strusafe.2008.06.021, 2009.

Nicholls, R. J.: Impacts of and Responses to Sea-Level Rise, in: Understanding Sea-Level Rise and Variability, edited by Church, J. A., Woodworth, P. L., Aarup, T., and Wilson, W. S., pp. 17–51, Wiley -Blackwell, ISBN: 978-1-4443-3452-4, 2010.

Nicholls, R. J. and Cazenave, A.: Sea-Level Rise and Its Impact on Coastal Zones, Science, 328(5985), 1517–1520, https://doi.org/10.1126/science.1185782, 2010.

Olesen, O., Kierulf, H. P., Brönner, M., Dalsegg, E., Fredin, O., and Solbakk, T.: Deep weathering, neotectonics and strandflat formation in Nordland, northern Norway, Nor. J. Geol., 93, 189–213, 2013.

Ouassou, M., Jensen, A. B. O., Gjevestad, J. G. O., and Kristiansen, O.: Next Generation Network Real-Time Kinematic Interpolation Segment to Improve the User Accuracy, International Journal of Navigation and Observation, 2015, article ID 346 498, https://doi.org/10.1155/2015/346498, 2015.

Passeri, D. L., Hagen, S. C., Medeiros, S. C., Bilskie, M. V., Alizad, K., and Wang, D.: The dynamic effects of sea level rise on low-gradient coastal landscapes: A review, Earth's Future, 3, 159–181, https://doi.org/10.1002/2015EF000298, 2015.

Poulter, B. and Halpin, P. N.: Raster modelling of coastal flooding from sea-level rise, Int. J. Geogr. Inf. Sci., 22(2), 167–182, https://doi.org/10.1080/13658810701371858, 2008.

Ravndal, O. R. and Sande, B. H.: Ekstremverdianalyse av vannstandsdata langs norskekysten (in Norwegian), Tech. rep., Norwegian Mapping Authority, Hydrographic Service, NDDF 16-1, 2016.

Reutebuch, S. E., McGaughey, R. J., Andersen, H. E., and Carson, W. W.: Accuracy of a high-resolution lidar terrain model under a conifer forest canopy, Can. J. Remote. Sens., 29(5), 527–535, https://doi.org/10.5589/m03-022, 2003.

Roelvink, D., Reniers, A., Van Dongeren, A., van Thiel de Vries, J., McCall, R., and Lescinski, J.: Modelling storm impacts on beaches, dunes and barrier islands, Coast. Eng., 56, 1133–1152, https://doi.org/10.1016/j.coastaleng.2009.08.006, 2009.

Rowley, R. J., Kostelnick, J. C., Braaten, D., Li, X., and Meisel, J.: Risk of rising sea level to population and land area, Eos, Transactions, American Geophysical Union, 88(9), 105–107, 2007.

Sibson, R.: A brief description of natural neighbor interpolation, in: Interpreting Multivariate Data, edited by Barnett, V., chap. 2, pp. 21–36, John Wiley, ISBN 978-047128039, 1981.

[revised manuscript text omitted]

Soft sand dunes at Sandestranda close to Randaberg. Photo: Oda R. Ravndal.

[Figure]

**Figure 7.** Stack of inundation maps indicating areas affected by coastal flooding in Bergen. Violet: Present MHW. Green: MHW at 2090 (0.71 m above present MHW). Orange: present 200-year storm surge (0.96 m above present MHW). Red: 200-year storm surge for 2090 (1.68 m above present MHW).

[Figure]

**Figure 8.** The bars indicate the size of areas (left), the number of buildings (middle), and the length of roads (right) affected by sea-level rise and storm surge in Norway. For each water level, the left and right bars indicate affected objects at present and for 2090, respectively. Percentages on top of right bars indicate total's change from now to 2090 due to sea-level rise.

[Figure]

**Figure 9.** Affected land areas due to a 200 year storm-surge hazard at present (left figures) , buildings and for 2090 (middleright figures),. The radius of the bubbles is for each municipality proportional to the size of flooded land area (upper figures) and roads flooded land areas as percentage of the municipality's total area (rightlower figures).

[Figure]

**Figure 10.** Affected buildings due to a 200 year storm-surge hazard at present (left) and for 2090 (right). The radius of the bubbles are proportional to the  number of exposed buildings.

[Figure]

**Figure 11.** Affected roads due to a 200 year storm-surge hazard at present (left) and for 2090 (right). The radius of the bubbles are proportional to the length of exposed roads.

[Figure]

**Figure 12.** The ten municipalities with most land-areas, buildings, and roads affected by a 200-year storm-surge hazard at present sea level.

[Figure]

**Figure 13.** Similar as Figure 12, but for 2090. The percentages indicate total's change from now to 2090 due to sea-level rise.

**High accuracy coastal flood mapping for Norway using LiDAR data - Reply to reviewer 1**

Kristian Breili[1,2], Matthew James Ross Simpson[1], Erlend Klokkervold[3], and Oda Roaldsdotter Ravndal[4]

[1]Geodetic Institute, Norwegian Mapping Authority, 3507 Hønefoss, Norway
[2]Faculty of Science and Technology, Norwegian University of Life Sciences, 1432 Ås, Norway
[3]Geographic Information System Development, Norwegian Mapping Authority, 3507 Hønefoss, Norway
[4]Hydrographic Service, Norwegian Mapping Authority, 4021 Stavanger, Norway

**Correspondence:** Kristian Breili (kristian.breili@kartverket.no)

**Note: All line and figure numbers refer to submitted manuscript**

**General comments**

This is a generally well-written, informative description of a new dataset and suite of tools for coastal management activities for the country of Norway. Thank you for affording me the opportunity to review! The analysis is thorough and description of input data, results, and related uncertainties is sufficient. I, however, would prefer to see this presented as a brief communication, as I believe the major contribution of this work lies more in its presentation of the dataset and availability of management tools and less so in any analyses of coastal risk, adaptations, or impacts. For example, I feel like the conclusions section (L450 – 473) can mostly represent the results section without any significant loss of interpretative value, and the remaining lines (to L493) appropriately represents sufficient discussion of uncertainties and overarching themes.

This opinion is not strong enough, however, to merit a major revision or rejection. I believe it's still an important contribution. Additionally, the assigned editor has looked over the manuscript and deemed it appropriate as a research article, so I will defer to their judgement here.

**Author comment:** Firstly, many thanks for your review which has improved the quality of our manuscript. We don't agree that the work should be presented as a short communication. It is certainly true that one of the main contributions of this work lies in the presentation of the dataset and availability of online management tools. However, there are also important results showing Norway's vulnerability to coastal flooding; including specific examples, regional differences, and detailing what is at risk. There is also an important assessment of the DEM accuracy. We feel there is more than enough analysis here to warrant keeping this paper as a research article. Reducing the work to a short communication (2-4 pages) would mean so much of the analysis would be lost. Furthermore, much of this analysis and the results go above and beyond what is available from our online tool SeHavnivå. It is also the case that this work aims to document Norway's present vulnerability to coastal flooding. As this work is, therefore, an important benchmark. As new knowledge and data sets become available then the numbers of what is at risk will change - including what is available online. We will keep

this work as a research article, as long as the editor does not require a complete rewriting of the manuscript to a brief communication.

**Other general notes**

**For the uncertainty analyses**

- With confidence intervals (or upper/lower limits of uncertainty) expressed for 9 of 10 metrics in Table 7, I'm not sure why there was no utilization of these in the results. At the least, one figure could have shown the difference in inundation extent using an upper and lower limit, and at most, all results could have been expressed as their appropriate ranges incorporating all relevant uncertainties

  **Author comment:** This is a preliminary assessment of the uncertainties - the purpose of which is to indicate that the uncertainties are generally smaller than the projected sea level rise. Also, to give the reader an idea of the different uncertainties involved in the mapping method. There are also smaller errors associated with the different vertical datums and transformations that have not been assessed for the entire coast.

  We agree that it would be very useful to show confidence intervals. However, as the other reviewer suggests, this could be highlighted as something to be addressed in future work. We therefore had added the following text to show that we will aim, in future, to perform a more complete uncertainty analysis, and to try and build that into our mapping.

  **Text added at line 369:** "In summary, a preliminary assessment indicates that the elevation model (RMSE 0.26 m) is the largest source of uncertainty in our mapping method. There are also smaller errors associated with different vertical datums and transformations between datums that have not been assessed for the entire coast. However, we believe that the sum of these mapping errors are generally smaller than the projected sea-level rise, which gives us confidence in our results. Future work should look at how these uncertainties can be incorporated into our mapping and web tool (Gesch, 2013, 2018; Cooper and Chen, 2013; Cooper et al., 2013)."

- One of the greatest sources of uncertainty as discussed was the bias introduced by engineered structures over water. I'm curious as to why the authors did not attempt to crop out these structures using a coastline mask. Was it because of discussed issues with the coastline not agreeing between datasets?

  **Author comment:** This is a problem we have struggled with but unfortunately there is no obvious solution. Lines 266 onwards already explain the issue: "We can not, however, simply subtract the numbers calculated for present MHW from the numbers for higher water levels for buildings, because an unknown number of these buildings will truly be affected by higher levels of flooding. We suggest that the numbers of buildings erroneously mapped as affected will decrease for higher water levels. The numbers calculated for present MHW for buildings form a basis estimates for other water levels can be compared to. They can also be considered as a measure of the precision of

the current methods and data used in our analysis. Note that because the coastal climate service Sehavnivå i kart presents numbers including the MHW-bias, the numbers for affected areas and roads given in Table 2 and 4 will differ from those of Se havnivå i kart."

Using a coastline to crop out engineered structures over water will not solve this problem. We do not know the height of these structures, and, as explained in lines 266 onwards, many of them will be affected by coastal flooding for water levels above MHW. In our work, we have chosen to include structures over water in the numbers for present MHW, for reasons explained above. But it is clearly something the reader should keep in mind when interpreting the results.

- A DEM accuracy of <0.1m (Table 7) is not really true, is it? That refers to the accuracy of the processed lidar point data, and not the interpolated DEM. As stated in the text, an accuracy of 0.26m is more appropriate. So why is it presented as such in the table, and elsewhere in the text?

  **Author comment:** Yes, it is correct that the project goal standard deviation applies to the laser data and not the interpolated DEM. To make this clearer, we have changed the order of Section 4.1 (deals with all uncertainties) and Section 4.2 (deals specifically with the DEM) and made several minor changes to the text to distinguish more clearly between accuracy of LiDAR data and interpolated DEM.

  **Updates to manuscript:**

  - Table 7: 0.26 m is added as DEM's estimated RMSE
  - Line 162: The vertical accuracy of the LiDAR data has a production goal root mean square error (RMSE) of 0.1 m for well-defined solid areas (Kartverket 2014).
  - Line 348: The project goal uncertainty (RMSE) of the LiDAR data, from which the DEM is interpolated from, is 0.1 m (Kartverket, 2014).
  - Line 351: The actual accuracy of the interpolated DEM depends on...
  - Line 356: We therefore consider the project goal uncertainty of the LiDAR data as an optimistic error estimate for the DEM in the coastal zone.
  - Line 406-407: Our tests suggest that the interpolated DEM used to calculate the inundation maps, does only achieve the project goal uncertainty of the LiDAR data in flat terrain. Considerably lower accuracies must be expected in steep areas and along much of the coast.

**For the figures:**

- I would like to have seen combination figures – pictures inset, side-by-side, or multi-panel with the representative inundation maps (e.g. 5 & 6, 7 & 8). Larger, too.

  **Author comment:** Figure 6 and 8 are now insets of Figure 5 and 7. Figure 4, 5, 7, and 9 will be two-column figures in order to make the maps larger and easier to read. Cross references and figure-captions are changed accordingly.

- Clear graphs showing the results from tables 2-6 more clearly would increase the impact of these findings. The bar graphs in Figures 10, 13, 14 are informative, but I can't help but feel like more data could be incorporated into larger line graphs for more interpretive power, and showcase the infrastructure challenges facing Norway in the RCP8.5 scenario.

  **Author comment:** We agree that more could be made of these graphs. To address this, we have changed figure 10 to include the percentage increase between present and 2090. Figure 13 and 14 now show that percentage of the total area of the municipality affected. Figure 14 also shows the percentage increase from present to 2090. See new figures in supplementary document.

- Figures 11/12 are hard to interpret, too much overlapping data. Perhaps colour-magnitude hexagons might more clearly convey the spatial patterns (e.g. see Figure 2, https://www.nature.com/articles/s41467-019-10762-4)

  **Author comment:** As long as the exact positions of the affected objects are not part of our data sets, we do not see how colour-magnitude hexagons can be used to map the impact of each municipality. So we have kept the bubbles, but made the figures larger and used small bubbles with a fixed size for municipalities with few structures affected. We have also reorganized the figures: in the revised manuscript Figure 11 and 12 are replaced by Figure 9, 10, and 11 that separately show the affected land areas, buildings, and roads for a 200-year storm surge at present and for 2090. This allows each theme to be visualized in larger figures and we believe they now are easier to interpret. See new figures in supplementary document.

- References are minimal, and several are non-peer reviewed reports. A more thorough examination of the literature, particularly with regards to inundation mapping and DEM analyses/uncertainties, would really benefit this manuscript

  **Author comment:** We have added the following references to address this issue:

    Section 2.1: Olesen et al. (2013)

    Section 4.1: Gesch (2013), Gesch (2018), Cooper and Chen (2013), Cooper et al. (2013)

    Section 2.2: Sibson (1981), Passeri et al. (2015), and Roelvink et al. (2009)

**Line by line comments as follows**

- L13 - Adaption and adaptation are used interchangeably throughout the manuscript. Please pick one for readability.

  **Author comment:** OK, adaption changed to adaptation throughout manuscript

- L19 - The sentence beginning "The consequences ..." is awkward. Maybe remove "and many" as well as the "the" before coastal.

  **Author comment:** OK, done.

- L24 - is GIA the only component of VLM at play? No tectonics? I ask because I don't know and a cursory look turned up no information. Just curious!

    **Author comment:** This is somewhat addressed on line 105: ". The VLM field used in the projections is based upon permanent GPS observations and repeated levelling (see Simpson et al., 2015). The presence of small-scale anomalies, e.g., urban subsidence or neotectonics (e.g. Olesen et al., 2013), may cause VLM to deviate significantly from this field at the local level."

    In other words, the VLM field is based on observations, it is entirely possible that a component of this motion is tectonic. Also, as stated, there will be local deviations from the field. However, the general pattern of vertical motion is clearly GIA. We have added a reference to neotectonics (Olesen et al., 2013) to address this a little more clearly.

- L78 - See point above, how this manuscript may be better suited as a short communication, describing the tool and its applications, inviting the reader to go and perform their own analyses.

    **Author comment:** See first comment.

- L128 - I recognize that it's an even more complicated variable than those already left out from your analyses, but I would have appreciated some mention of a reduction in sea ice and associated coastal effects on arctic communities. In Canada the highest rates of coastal retreat and impact on communities/infrastructure due to SLR is in arctic areas impacted by a loss of sea ice and associated increased wave activity. . . I'm sure there are some similar effects being witnessed in northern parts of the study area.

    **Author comment:** Mainland Norway is essentially free of sea ice. Although situated quite far north, the Gulf stream means the coast remains free of sea ice throughout the year.

- L161 - is that a definitive statement – every grid cell has at least 2 datapoints? Or is that an average. Also, interpolation method?

    **Author comment:** The product specifications of the LiDAR data says "at least two datapoints per $m^2$".

    **Text added at line 162:** "Two methods are applied to interpolate the LiDAR data to a regular DEM. In a first try, natural neighbor interpolation (Sibson, 1981) was used. If this failed, empty spaces were binned with an average value."

- L168 - Some discussion of alternatives to the bathtub methods (e.g. modeling approaches using XBeach)?

    **Added at line 169:** "The "bathtub" approach is favored for several reasons. Firstly, mapping results from this approach are consistent with how current guidelines on coastal planning are applied in Norway. Secondly, the approach is straightforward, computationally inexpensive, and has been widely used in large-scale coastal flooding

analyses. However, there are known limitations of the "bathtub" method. For example, the response of hydrodynamics, morphology, and ecology as sea level rises is not accounted for (see Passeri et al. (2015) for a review). Some of these effects could be important on local scales along the Norwegian coast."

**Added at line 440:** "The "bathtub" approach applied in the present study results in maps that are consistent with national guidelines on how to account for future sea-level change and storm-surge in coastal planning. Currently, there are no regulations for modelling the effects of waves, which may increase mean sea level during a storm and introduce geomorphological changes due to erosion and transport of sediments. Modelling the effects of waves should be addressed in future work and will require a more advanced framework than the "bathtub" approach. A more advanced framework is provided by the open-source numerical model XBeach (Roelvink et al., 2009), which is developed to simulate the effects of storms on sandy coasts with domain size of kilometers. The XBeach model is not a tool for analyzing the entire Norwegian coast, but is suitable for case studies of vulnerable areas like beaches and coasts covered by soft sediments (e.g., southwest of Norway)."

- L193 - Again, unless this is a short communication introducing the tool, this kind of discussion isn't really necessary in a scientific paper

  **Author comment:** Ok, removed

- L202 - See general comments for sections 3 and 4. Generally I think that sections 3.1 and 3.2 could be thinned significantly, particularly if more detailed and interpretive figures and maps are presented. 3.3 is a good section – this is the level of interpretation I'd personally like to see in the results.

  **Author comment:** We have moved line 251-272 (discussing the MHW bias) from Section 3.2 to Section 2.3. This makes Section 3.2 considerably shorter and more focused on the results. The figures have also been revised somewhat to include more information. However, we disagree that the level of interpretation is too detailed in Sections 3.1 and 3.2. Section 3.1 deals with specific examples, and the descriptions of those inundation maps are, we would argue, brief and light on detail. The purpose of these examples is to show how different geographic/landscape situations are at risk. And, furthermore, to show how Norway is at risk on local scales. Section 3.2 has been cut as the MHW-bias discussion has been removed. Otherwise we believe the level of interpretation is appropriate for this work.

- Sections 4.1 and 4.2 are great – but then none of these uncertainties so carefully outlined are included in the analysis!

  **Author comment:** Confidence intervals have not been calculated for the maps available in the coastal service, but future work should look at how these uncertainties can be incorporated into our mapping and web tool. See also comment above.

- L421 - "... and future applications of this tool" or something to that effect?

  **Author comment:** Ok, header changed to "Comparison to other studies and future work"

**Other corrections applied by the authors**

Table 1: The sub category is changed from "Private" to "Private industry" for "factories, workshops, storage halls, power plants, and transformers".

Table 5: Horizontal lines added between each category

Figure 3 has been improved. The new version is clearer and easier to interpret.

In the revised manuscript, we have replaced Figure 11 and 12 by Figure 9, 10, and 11 that separately show the affected land areas, buildings, and roads for a 200-year storm surge at present and for 2090. This allows each theme to be visualized in larger figures.

Line 52: objects of impact -> objects at risk

Line 72: line break added

Line 129: Reference updated from Nicholls and Cazenave (2010) to Bamber et al. (2018).

Line 179: correcting typos: ...the object's height is used determines whether...

Line 194 changed to: "Furthermore, the maps and numbers presented in Se havnivå i kart will be regularly updated as new knowledge and data (e.g. new elevation data, better understanding of vertical datums, error corrections) becomes available."

Line 203 changed to: "...the three types of coastlines (strandflat, glaciofluvial deltas, and soft moraine coast)..."

Change at line 313: "The municipalities with the largest land areas that are at risk of flooding are located in the middle of Norway (between Trondheim and Tromsø) and in the outer part of Oslofjorden."

Added at line 314: ...evident by the upper left...

Line 383: solid bedrock -> exposed bedrock

**Correspondence:** Kristian Breili (kristian.breili@kartverket.no)

**Note: All line and figure numbers refer to submitted manuscript**

This paper is an effective study of sea-level rise and coastal flooding exposure in Norway based on a high-quality DEM. It exhibits a sound, straightforward approach that uses many of the best practices that have been established for these types of coastal assessments. The paper documents well the data, methods used, and results, and the tables and figures effectively support the material in the text. Overall, the Discussion section is very good, especially the factors affecting uncertainty and the accuracy of the DEM.

**Author comment:** Many thanks for your review which has improved the quality of our manuscript.

The results could be improved by attaching confidence levels to the estimates of impacted area and objects. This would entail not just describing the accuracy of the DEM (and the associated datum conversions), but applying that cumulative vertical uncertainty to characterize the confidence of the results (see Gesch, 2013 and Gesch, 2018 for details on how this is done). All the needed information is already available with the comprehensive DEM accuracy assessment that has been done and all the other uncertainty information that is listed in Table 7. I am not saying that this needs to be done for this paper to be accepted, as I believe the results as currently presented are useful, but adding confidence information could be done in future related work (perhaps as the remaining 20% of the country is worked on and national results are revised and added to), and the authors could add this idea of characterizing the confidence of the results to the Discussion/Conclusions sections.

**Author comment:** This is definitely something we will address in future work.

**Added to discussion (Section 4.1):** "Future work should look at how these uncertainties can be incorporated into our mapping and web tool (Gesch, 2013, 2018; Cooper and Chen, 2013; Cooper et al., 2013)."

(See also reply to comment to line 334)

Comment on lines 207-216 (discussion of Smola) in section 3.1, and Figures 13 and 14 that it refers to: The area affected is important, but without knowing the total area of each of the ten municipalities (assuming there's variability in the

areas) it's hard to see which ones are affected the most. So you could also show the affected area as a percent of the total area of each municipality as a way to rank the municipalities.

**Author comment:** Thank you for this nice suggestion. The updates of Figure 11 and 12 (Figure 9 in revised manuscript) include the percentage affected total area of each municipality. See new figures in supplementary document.

**Added at line 216:** "Taking into account the municipalities total area, Smøla is the tenth and ninth most affected municipality by a 200 year storm at present and for 2090, respectively."

**Added at line 317:** "When considering flooded land area as a percentage of the total municipality area (lower panels of Figure 9), we find that six of the ten municipalities with the highest percentages are located on the west coast."

Line 3 (abstract): What about isostatic rebound? I see it is mentioned in lines 24-28 as being important, so may want to also mention it here

**Added to abstract:** "...and land uplift due to glacial isostatic adjustment"

Line 50: Suggest "potential consequences" here

**Author comment:** Ok, "potential" added

Line 81: It is good that the DEM used has high accuracy (0.26 m RMSE) so that these 1 m intervals can be effectively mapped at high confidence

**Author comment:** Yes, this point is mentioned in line 154. We have added a reference to Gesch (2018).

Line 110: Given the accuracy of the DEM used in this study (0.26 m RMSE), each of these will be mapped with a different level of confidence. 0.40 m is closer to the inherent noise level of the DEM, so will be mapped with less confidence than 0.82 m. For further information see Gesch, 2018.

**Author comment:** This is an important remark.

**Text added to Section 4.1, line 347:** "Given the accuracy of the DEM used in this study (0.26 m RMSE), each water level will be mapped with a different level of confidence because the lower levels are close to the inherent noise level of the DEM."

Line 172: Although these disconnected low-lying areas can be important to account for too, especially for storm surge flooding that may overtop the barriers because of waves, or rising groundwater due to sea-level rise. Some studies have mapped and reported these areas separately. See Rotzoll and Fletcher, 2012; Copper et al., 2013; Copper et al., 2015).

**Author comment:** Thank you for making us aware of this. We have decided not to comment on this in the text because this will not be a significant problem for our coastal service due to the generally steep topography of Norway.

Line 230: "region's"

**Author comment:** OK, corrected

Line 275: This sound like a key finding from this study

**Text added to Conclusions at line 457:** "Notably, we also find that the numbers of affected objects for a 20 year storm-surge return height in 2090 will exceed the numbers for the 1000 year storm-surge at present. Indicating that an increasing number of objects will be at risk of more frequent flooding."

Line 304: This, of course, assumes that no protective structures are build (seawals, levees, etc.)

**Text changed to:** "In this scenario, more than 1700 km$^2$, 263,000 buildings, and 6800 km of roads would be permanently flooded if no adaptive measures are taken."

Line 334: But the uncertainties could be accounted for. See: Gesch, 2013; Gesch 2018; Cooper and Chen, 2013; Cooper et al., 2013.

**Author comment:** This is a preliminary assessment of the uncertainties - the purpose of which is to indicate that the uncertainties are generally smaller than the projected sea-level rise. Also, to give the reader an idea of the different uncertainties involved in the mapping method. There are also smaller errors associated with the different vertical datums and transformations that have not been assessed for the entire coast.

We agree that it would be very useful to show confidence intervals. However, we have not come so far in our work that we at present can build that into our mapping.

**Added to end of Section 4.1:** "In summary, a preliminary assessment indicates that the elevation model (RMSE 0.26 m) is the largest source of uncertainty in our mapping method. There are also smaller errors associated with different vertical datums and transformations between datums that have not been assessed for the entire coast. However, we believe that the sum of these mapping errors are generally smaller than the projected sea-level rise, which gives us confidence in our results. Future work should look at how these uncertainties can be incorporated into our mapping and web tool (Gesch, 2013, 2018; Cooper and Chen, 2013; Cooper et al., 2013)."

Line 370: The correct name is "National Geodetic Survey".

**Author comment:** Ok, corrected.

Figure 1: These numbers are meters, right? It should indicate that.

**Text added to caption:** "For all figures the unit is meter."

**Other corrections applied by the authors**

Table 1: The sub category is changed from "Private" to "Private industry" for "factories, workshops, storage halls, power plants, and transformers".

Table 5: Horizontal lines added between each category

85 Figure 3 has been improved. The new version is clearer and easier to interpret.

In the revised manuscript, we have replaced Figure 11 and 12 by Figure 9, 10, and 11 that separately show the affected land areas, buildings, and roads for a 200-year storm surge at present and for 2090. This allows each theme to be visualized in larger figures.

Line 52: objects of impact -> objects at risk

90 Line 72: line break added

Line 129: Reference updated from Nicholls and Cazenave (2010) to Bamber et al. (2018).

Line 179: correcting typos: ...the object's height is used determines whether...

Line 194 changed to: "Furthermore, the maps and numbers presented in Se havnivå i kart will be regularly updated as new knowledge and data (e.g. new elevation data, better understanding of vertical datums, error corrections) becomes

95 available."

Line 203 changed to: "...the three types of coastlines (strandflat, glaciofluvial deltas, and soft moraine coast)..."

Change at line 313: "The municipalities with the largest land areas that are at risk of flooding are located in the middle of Norway (between Trondheim and Lofoten Tromsø) and in the outer part of Oslofjorden."

Added at line 314: ...evident by the upper left...

100 Line 383: solid bedrock -> exposed bedrock

**Correspondence:** Kristian Breili (kristian.breili@kartverket.no)

**Note: All line and figure numbers refer to originally submitted manuscript**

**Changes to text**

- Adaption changed to adaptation throughout manuscript

- L3, text added "...steep topography and land uplift due to glacial isostatic adjustment"

- L19, changed: "The consequences of increasing sea level are large because coastal zones are densely populated areas, have a large population growth, and are economically important.

- L50: "potential" added

- L52: objects of impact -> objects at risk

- L72, line break added

- L108, text added: "or neotectonics (Olesen et al., 2013)"

- L129, reference updated from Nicholls and Cazenave (2010) to Bamber et al. (2018).

- L155, reference to Gesch (2018) added.

- L162, text added: "Two methods are applied to interpolate the LiDAR data to a regular DEM. In a first try, natural neighbor interpolation (Sibson, 1981) was used. If this failed, empty spaces were binned with an average value."

- L162, canged: The vertical accuracy of the LiDAR data has a production goal root mean square error (RMSE) of 0.1 m for well-defined solid areas (Kartverket 2014).

- L169, text added: "The "bathtub" approach is favored for several reasons. Firstly, mapping results from this approach are consistent with how current guidelines on coastal planning are applied in Norway. Secondly, the approach is straightforward, computationally inexpensive, and has been widely used in large-scale coastal flooding analyses. However, there are known limitations of the "bathtub" method. For example, the response of hydrodynamics, morphology, and ecology as sea level rises is not accounted for (see Passeri et al. (2015) for a review). Some of these effects could be important on local scales along the Norwegian coast."

- L179, typos corrected: "...the object's height is used determines whether..." -> "...the object's height determines..."

- L190, text removed: "e.g., buildings on piers and roads on bridges (see Figure 2)"

- L192: L251-L254) from Section 3.2 inserted.

- L193, sentence removed: The service's web client does not process data on the fly. All map layers and statistics are preprocessed and read from a database in order to ensure a smooth user experience.)

- L193: L254-L272 from Section 3.2 inserted.

- L194, changed to: "Furthermore, the maps and numbers presented in Se havnivå i kart will be regularly updated as new knowledge and data (e.g. new elevation data, better understanding of vertical datums, error corrections) become available."

- L203, changed to: "...the three types of coastlines (strandflat, glaciofluvial deltas, and soft moraine coast)..."

- L216, text added: "Taking into account the municipalities total area, Smøla is the tenth and ninth most affected municipality by a 200 year storm at present and for 2090, respectively."

- L230, correction: "regions" -> "region's"

- L251-L272 moved from Section 3.2 to Section 2.3.

- L313, changed: "The municipalities with the largest land areas that are at risk of flooding are located in the middle of Norway (between Trondheim and Tromsø) and in the outer part of Oslofjorden."

- L304, text added: "...if no adaptive measures are taken."

- L313, changed: "between Trondheim and Lofoten" -> " between Trondheim and Tromsø"

- L314, text added: ...evident by the upper left...

- L317, text added: "When considering flooded land area as a percentage of the total municipality area (lower panels of Figure 9), we find that six of the ten municipalities with the highest percentages are located on the west coast."

- Section 4.1 and 4.2: We have changed the order of Section 4.1 (deals with all uncertainties) and Section 4.2 (deals specifically with the DEM) and made several minor changes to the text to distinguish more clearly between accuracy of LiDAR data and interpolated DEM.

- L347, text added: "Given the accuracy of the DEM used in this study (0.26 m RMSE), each water level will be mapped with a different level of confidence because the lower levels are close to the inherent noise level of the DEM."

- L348, changed: The project goal uncertainty (RMSE) of the LiDAR data, from which the DEM is interpolated from, is 0.1 m (Kartverket, 2014).

- L351, changed: The actual accuracy of the interpolated DEM depends on...

- L356, changed: We therefore consider the project goal uncertainty of the LiDAR data as an optimistic error estimate for the DEM in the coastal zone.

- L369, text added "In summary, a preliminary assessment indicates that the elevation model (RMSE 0.26 m) is the largest source of uncertainty in our mapping method. There are also smaller errors associated with different vertical datums and transformations between datums that have not been assessed for the entire coast. However, we believe that the sum of these mapping errors are generally smaller than the projected sea-level rise, which gives us confidence in our results. Future work should look at how these uncertainties can be incorporated into our mapping and web tool (Gesch, 2013, 2018; Cooper and Chen, 2013; Cooper et al., 2013)."

- L373, correction: "National Geological Survey" -> "National Geodetic Survey".

- L383, changed: "solid bedrock" -> "exposed bedrock"

- L406-407, changed: Our tests suggest that the interpolated DEM used to calculate the inundation maps, does only achieve the project goal uncertainty of the LiDAR data in flat terrain. Considerably lower accuracies must be expected in steep areas and along much of the coast.

- L421, header of section changed to "Comparison to other studies and future work"

- L440, text added: "The "bathtub" approach applied in the present study results in maps that are consistent with national guidelines on how to account for future sea-level change and storm-surge in coastal planning. Currently, there are no regulations for modelling the effects of waves, which may increase mean sea level during a storm and introduce geomorphological changes due to erosion and transport of sediments. Modelling the effects of waves should be addressed in future work and will require a more advanced framework than the "bathtub" approach. A more advanced framework is provided by the open-source numerical model XBeach (Roelvink et al., 2009), which is developed to simulate the effects of storms on sandy coasts with domain size of kilometers. The XBeach model is not a tool for analyzing the entire Norwegian coast, but is suitable for case studies of vulnerable areas like beaches and coasts covered by soft sediments (e.g., southwest of Norway)."

- L457, text added: "Notably, we also find that the numbers of affected objects for a 20 year storm-surge return height in 2090 will exceed the numbers for the 1000 year storm-surge at present. Indicating that an increasing number of objects will be at risk of more frequent flooding."

- Table 1: The sub category is changed from "Private" to "Private industry" for "factories, workshops, storage halls, power plants, and transformers".

- Table 5: Horizontal lines added between each category

- Table 7: 0.26 m is added as DEM's estimated RMSE

- Added references:

  Section 2.1: Olesen et al. (2013)

  Section 2.2: Sibson (1981), Passeri et al. (2015), and Roelvink et al. (2009)

  Section 4.1: Gesch (2013), Gesch (2018), Cooper and Chen (2013), Cooper et al. (2013)

**Changes to figures**

- Caption Figure 1, text added: "For all figures the unit is meter."

- Figure 3 has been improved. The new version is clearer and easier to

- Figure 6 and 8 are now insets of Figure 5 and 7.

- Figure 4, 5, 7, and 9 are changed to two-column figures in order to make the maps larger and easier to read. Cross references and figure-captions are changed accordingly.

- We have changed Figure 10 to include the percentage increase between present and 2090. Figure 13 and 14 now show the percentage of the total area of the municipality affected. Figure 14 also shows the percentage increase from present to 2090.

- Figure 11/12: The figures are made larger and small bubbles with a fixed size are used for municipalities with few structures affected. We have also reorganized the figures: in the revised manuscript Figure 11 and 12 are replaced by Figure 9, 10, and 11 that separately show the affected land areas, buildings, and roads for a 200-year storm surge at present and for 2090. This allows each theme to be visualized in larger figures and we believe they now are easier to interpret. interpret.